# *In silico* identification of sugarcane (*Saccharum officinarum* L.) genome encoded microRNAs targeting sugarcane bacilliform virus

**Muhammad Aleem Ashraf**[1,2]☯*, **Xiaoyan Feng**[1]☯, **Xiaowen Hu**[3]☯, **Fakiha Ashraf**[1], **Linbo Shen**[1], **Muhammad Shahzad Iqbal**[4], **Shuzhen Zhang**[1]*

**1** Institute of Tropical Bioscience and Biotechnology, Sugarcane Research Centre of Chinese Academy of Tropical Agricultural Sciences, Haikou, China, **2** Department of Bioscience and Technology, Khwaja Fareed University of Engineering and Information Technology, Rahim Yar Khan, Pakistan, **3** Zhanjiang Experimental Station, Chinese Academy of Tropical Agricultural Sciences, Zhanjiang, China, **4** Department of Biotechnology, University of Central Punjab, Lahore, Pakistan

☯ These authors contributed equally to this work.
* aleem.ashraf@kfueit.edu.pk (MAA); zhangshuzhen@itbb.org.cn (SZ)

**Data Availability Statement:** All relevant data are within the manuscript and its Supporting Information files.

## Abstract

Sugarcane bacilliform virus (SCBV) is considered one of the most economically damaging pathogens for sugarcane production worldwide. Three open reading frames (ORFs) are characterized in the circular, ds-DNA genome of the SCBV; these encode for a hypothetical protein (ORF1), a DNA binding protein (ORF2), and a polyprotein (ORF3). A comprehensive evaluation of sugarcane (*Saccharum officinarum* L.) miRNAs for the silencing of the SCBV genome using *in silico* algorithms were carried out in the present study using mature sugarcane miRNAs. miRNAs of sugarcane are retrieved from the miRBase database and assessed in terms of hybridization with the SCBV genome. A total of 14 potential candidate miRNAs from sugarcane were screened out by all used algorithms used for the silencing of SCBV. The consensus of three algorithms predicted the hybridization site of sof-miR159e at common locus 5534. miRNA–mRNA interactions were estimated by computing the free-energy of the miRNA–mRNA duplex using the RNAcofold algorithm. A regulatory network of predicted candidate miRNAs of sugarcane with SCBV—ORFs, generated using Circos—is used to identify novel targets. The predicted data provide useful information for the development of SCBV-resistant sugarcane plants.

## 1. Introduction

Sugarcane bacilliform viruses (SCBVs) are classified into the *Badnavirus* genus of the *Caulimoviridae* family. These viruses are composed of monopartite, circular, non-enveloped bacilliforms that are (30 × 120–150 nm) in size, with a double-stranded DNA (ds-DNA)- genome of approximately 7.2–9.2 Kbp in size [1]. The genome of SCBV constitutes three major open reading frames (ORFs) that are located on the 'plus DNA strand' with a single discontinuity [2]. ORF1 encodes a small hypothetical protein. ORF2 encodes a virion-associated DNA-

**Funding:** This work was supported by the Central Public-interest Scientific Institution Basal Research Fund for Chinese Academy of Tropical Agricultural Sciences (Grant number: 19CXTD-33), National Natural Science Foundation of China (Grant number: 31771865), the Sugar Crop Research System (Grant ID: CARS-170301) and the Talented Young Scientist Program of China (Grant ID: Pakistan-18-004). The funders had no role in the design of the study; in the collection, analyses, or interpretation of data, in the writing of the manuscript, or in the decision to publish the results.

**Competing interests:** The authors declare no conflict of interest.

binding protein. ORF3 encodes the largest polyprotein, represented as P3 here, and is composed of multiple functional sub-units. The polyprotein (P3) is cleaved by a viral aspartic protease to give rise to multiple functional small proteins, thereby forming intracellular movement, capsids, aspartic proteases, reverse transcriptase (RT), and ribonuclease H (RNase H) [1–6]. The RT-RNaseH-coding region is considered to be the most common taxonomic marker for the identification of badnaviral genomic components. This coding region is a standard source to compare the sequence diversity of the badnaviral genomes [7].

The first report of SCBV infection was observed in the Cuban sugarcane cultivar B34104 in 1985 [8]. These viruses have been disseminated worldwide and have reduced crop production significantly because of the accessibility and exchange of biological materials globally. SCBV is a source of infection for several bioenergy crop sugarcane cultivars, varieties, and species. The broad host range of the SCBV includes diverse and economically important members of the *Poaceae* (sugarcane, and rice) and *Musaceae* (banana) families. Natural transmission of SCBV is disseminated by sap-feeding mealybug species via vegetative cutting [9]. SCBV disease symptoms include chlorosis and leaf freckling. Infected sugarcane plants have also been monitored and feature no symptoms. In recent years in China, SCBV-infection in sugarcane plants has resulted in a reduced sucrose content, juice, stalk weight, purity, and gravity [6].

RNA silencing is an evolutionary conserved homolog-dependent regulatory mechanism of gene expression in all eukaryotes and is triggered by small RNA molecules (sRNA). dsRNA is the ultimate trigger of the RNAi complex, which works as a replication intermediate created by viral RNA-dependent RNA polymerases (RDRs) [10]. The RNAi mechanism works with cleavage of the precursor dsRNA into short 21–24 nt siRNA or miRNA duplexes using an RNaseIII-like enzyme called Dicer (DCL) [11–13].

The RNAi-mediated response of plants against invading viruses is especially significant during the infection period [14]. The RNAi mechanism inhibits protein translation at the mRNA level via a highly sequence-specific strategy [15]. Sugarcane has inherited an active immunity, consisting of small non-coding microRNAs (miRNAs) to control viral diseases. miRNA-mediated gene silencing is considered to validate the activity of positive or negative immune-based regulation; it is also considered a key activator of immune defense in plants [16, 17]. RNA silencing in the form of miRNAs within the host plant is a source of natural immunity. Such a mechanism provides resistance to the host plant after infection via foreign genetic elements, including plant viruses [18–20].

Artificial microRNA (amiRNA)-mediated RNAi produces a single 21-nucleotide amiRNA (analogous to a single siRNA) that only recognizes a target sequence with less than five mismatches. This feature not only ensures a higher silencing specificity for amiRNAs than hairpin RNAs but also offers unique advantages [21, 22]. amiRNA-mediated silencing of invading viruses in plants was first reported by Niu [23]. This amiRNA-based silencing strategy has been applied to with many plants in order to combat plant viruses, such as cotton leaf curl Kokhran virus (CLCuKoV) [24], cucumber mosaic virus (CMV) [25], cymbidium mosaic virus (CymMV), and odontoglossum ringspot virus (ORSV) [26].

In this study, we performed a comprehensive bioinformatics analysis to identify sugarcane miRNAs predicted to target the SCBV genome. Computational methods can determine how miRNAs target a desirable mRNA. A large number of computational algorithms are publicly available for miRNA target prediction. It is highly advantageous to acquire several computational tools with different features. Researchers are challenged with an important choice regarding selecting suitable tools for prediction [27]. The current study implements miRNA prediction algorithms and identifies potential targets of sugarcane-derived miRNAs against SCBV as a precedent for creating resistance in sugarcane cultivars using RNAi technology. Potential sugarcane miRNAs are also screened for understanding sugarcane–*Badnavirus*

interactions. The novel computational approach here supports the idea of generating SCBV-resistant sugarcane plants through genetic engineering.

## 2. Materials and methods

### 2.1. Retrieval of sugarcane MicroRNAs

Mature sugarcane microRNAs (miRNAs) and stem-loop hairpin precursor sequences were retrieved from the miRNA biological sequence database miRBase (v22) (http://mirbase.org/). miRBase serves as primary public repository and standard online reference resource for all published miRNA sequences, along with providing textual annotations and gene nomenclature [28–30]. In this study, 16 *S.officinarum* (MI0001756-MI0001769) and 19 Saccharum spp. (MI0018180- MI18197) miRNA sequences were downloaded (S1 Table in S1 File).

### 2.2. SCBV genome retrieval and annotation

The full-length transcript of the SCBV-BRU genome was isolated from the *S. officinarum* cultivar and then published, and available via accession no. JN377537 [31]. The expected sizes and abundances of the ORFs along nucleotide distributions of the above mentioned NCBI retrieved SCBV-BRU genome were estimated using the pDRAW32 DNA analysis software (version 1.1.129) (AcaClone software). The SCBV-BRU genome annotation represents ORFs of varying lengths.

### 2.3. Target prediction in SCBV genome

Target prediction is considered a key feature towards the identification of credible miRNA–mRNA interaction hybridization. At present, many target prediction algorithms have been designed to predict and identify the best miRNA target candidate. Each tool uses specific criteria and methods for miRNA target prediction. We used four target prediction algorithms cited in the literature (miRanda, RNA22, RNAhybrid and psRNATarget) to find the most relevant sugarcane miRNAs for silencing of the SCBV genome (Table 1). These computational tools compute the complementarity-based attachment of miRNA-mRNA. This attachment is divided into seed and mid regions. The mismatch in the seed region is more damaging than that of a mismatch in the middle region of miRNA-mRNA attachment. This provides the basis for over-sensitivity for the computation. We can set higher penalty of a mismatch in seed region which will make the prediction more sensitive. We designed an effective computational approach to analyze miRNA targets at three different prediction levels namely the individual, union, and intersection levels. A detailed workflow pipeline is presented in (Fig 1) below.

**Table 1. Comparison of distinctive parameters used in the common target prediction tools.**

| Tools | Algorithms | Seed pairing | Target site accessibility | Multiple sites | Translation Inhibition | Availability |
|---|---|---|---|---|---|---|
| miRanda | Local alignment | + | + | + | + | Web server and source code |
| RNA22 | FASTA | _ | + | + | _ | Only web server |
| RNAhybrid | Interamolecular hybridization | + | + | + | + | Web server and source code |
| psRNATarget | Smith-Waterman | _ | + | + | + | Only web server |
| Tapirhybrid | FASTA | + | + | + | _ | Web server and source code |
| Targetfinder | FASTA | + | _ | _ | _ | Only source code |
| Target-align | Smith-Waterman | _ | _ | + | _ | Web server and source code |
| Targetscan | Custom made | + | _ | + | + | Only source code |

'+' Represents a feature was used, '-'indicates that a feature was not used.

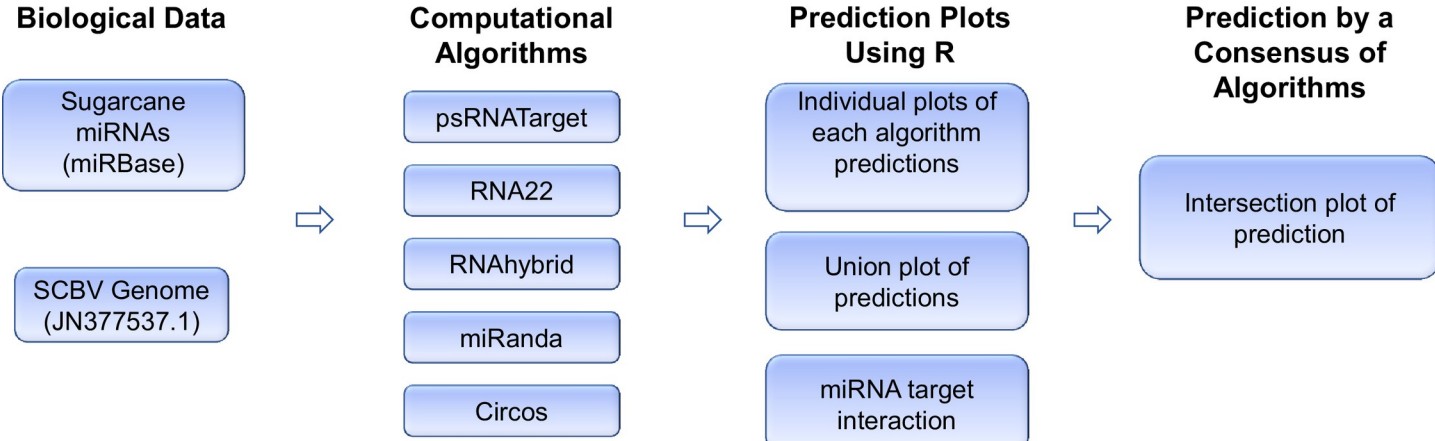

**Fig 1. The methodology of host or sugarcane miRNA target prediction in the SCBV genome.** A flowchart designed for predicting candidate miRNAs of host that could potentially target SCBV genome. The biological data are composed of sugarcane miRNAs retrieved from the miRBase database and SCBV genome from NCBI GenBank database. The algorithmic framework consists of three kinds of tools used for identification of sugarcane-encoded miRNA targets, prediction of secondary structures and visualization of mi RNA–target interaction. The R language was used to create plots and select data using in-house scripts/codes.

### 2.4. miRanda

miRanda is considered to be a standard miRNA–target predictor scanning algorithm. It was implemented for the first time in 2003 [32] and has been updated into a web-based tool for miRNA analysis [33]. The latest version of the miRanda software was accessed using the online source website (http://www.microrna.org/).

### 2.5. RNA22

RNA22 is a user-friendly, web-based (http://cm.jefferson.edu/rna22v1.0/) novel pattern-recognition algorithm that is used for predicting target sites with corresponding hetero-duplexes. Non-seed- based interaction, pattern recognition, site complementarity, and folding energy are the key parameters of the RNA22 algorithm [34]. Final scoring removes the need to use a cross-species conservation sequence filter [35].

### 2.6. RNAhybrid

RNAhybrid is an easy-to-use, fast, flexible, web-based (http://bibiserv.techfak.uni-bielefeld.de/rnahybrid) intermolecular hybridization algorithm that is used to estimate mi RNA–mRNA interaction as well perform target prediction based on MFE hybridization. A p-value is assigned to assess RNA–RNA interaction-based hybridization sites in the 3′ UTR sequence [36]. RNAhybrid is widely used to estimate the MFE of the consensual mi RNA–target pair and the mode of target inhibition as suggested [37].

### 2.7. psRNATarget

psRNATarget is a new web server (http://plantgrn.noble.org/psRNATarget/) that is used to identify the target genes of plant miRNAs based on a complementary matching scoring schema. It has been used to discover validated mi RNA–mRNA interactions [38]. The plant psRNATarget was designed to integrate a key function for miRNA target prediction using complementarity scoring and secondary structure prediction [39]. Target site accessibility was evaluated by estimating the unpaired energy (UPE) to unfold a secondary structure [37].

## 2.8. Mapping of mi RNA–target interaction

An interaction map was created between sugarcane miRNAs and SCBV ORFs using the Circos algorithm [40].

## 2.9. RNAfold

RNAfold is a new web-based algorithm and was applied for the prediction of the stable secondary structures of pre-miRNAs based on the MFEs [41].

## 2.10. Free energy (ΔG) estimation of duplex binding

RNAcofold is a novel web-based server (http://rna.tbi.univie.ac.at/cgi-bin/RNAWebSuite/RNAcofold.cgi) that is used for estimating free energy (ΔG) associated with miRNA–mRNA interactions [42]. The free energy of miRNA–miRNA duplexes is considered a key predictor for miRNA targeting during hybridization.

**2.10.1. *In Silico* sugarcane miRNA expression profiling.**   Plant miRNA Expression Atlas (PmiRExAt) is a web-based resource comprising a miRNA expression profile and searching tool for 1,859 wheat, 2,330 rice, and 283 maize miRNAs [43]. PmiRExAt can be accessed at http://pmirexat.nabi.res.in/. The sequences of mature microRNAs from sugarcane were blasted in PmiRExAt and the expression patterns of the homologous microRNAs were searched in wheat, rice, and maize.

**2.10.2. Graphical representation.**   All the computational data were processed into graphical representations using R version 3.1.1 [44].

# 3. Results

## 3.1. Genome Organization of SCBV

SCBV is a plant pararetrovirus that is, classified in the genus *Badnavirus* of the family *Caulimoviridae*. The genomic ds-DNA molecule of SCBV is comprised of three ORFs, separated by an intergenic region (IR). ORF1 is composed of 557 nucleotides (618–1175 nt), encoding a hypothetical protein (P1) with 185 amino acids (aa),while ORF2 is composed of 370 nucleotides (1176–1546 nt) codes for a virion-associated DNA binding protein (P2) with 123 aa. The precise functional capabilities of these proteins (encoded by ORF1 and ORF2) have not been explored. A large polyprotein (1977 amino acids) is encoded by ORF3 (1547–7479 nt) to cleave by a viral aspartic protease. The resulting proteins obtained are named as movement, capsid protein, aspartyl proteinase, reverse transcriptase and ribonuclease H. The IR is composed of 1022 nucleotides (7479–618) and is located between 3'-ORF3 to 5'-ORF1. The intergenic region (IR) works as a promoter and controls the transcription and regulation of the SCBV genome. The genome organization of the SCBV with three ORFs is shown in (Fig 2).

## 3.2. ORF1-encoding hypothetical protein

The hypothetical protein of the SCBV genome that is encoded by ORF1 had an unknown function [2]. In miRanda that only predicted two target sites for sugarcane miRNAs sof-miR156 and sof-miR168 at nucleotide positions 818–837 and 617–638 to target ORF1 (Fig 3A). RNA22 predicted the binding sites of miRNAs sof-miR156 and sof-miR168a at the two different locus positions of 817 and 834, respectively (Fig 3B). The RNAhybrid algorithm predicted multiple potential binding sites of sugarcane miRNAs sof-miR168 (a, b), ssp-mi827, and ssp-miR1128 at nucleotide positions 612–632, 1170–1192, and 1137–1157 respectively (Fig 3C). In addition, psRNATarget identified potential hybridization sites of sof-miR159 (c, e) at locus positions 1003 and 820 respectively (Fig 3D).

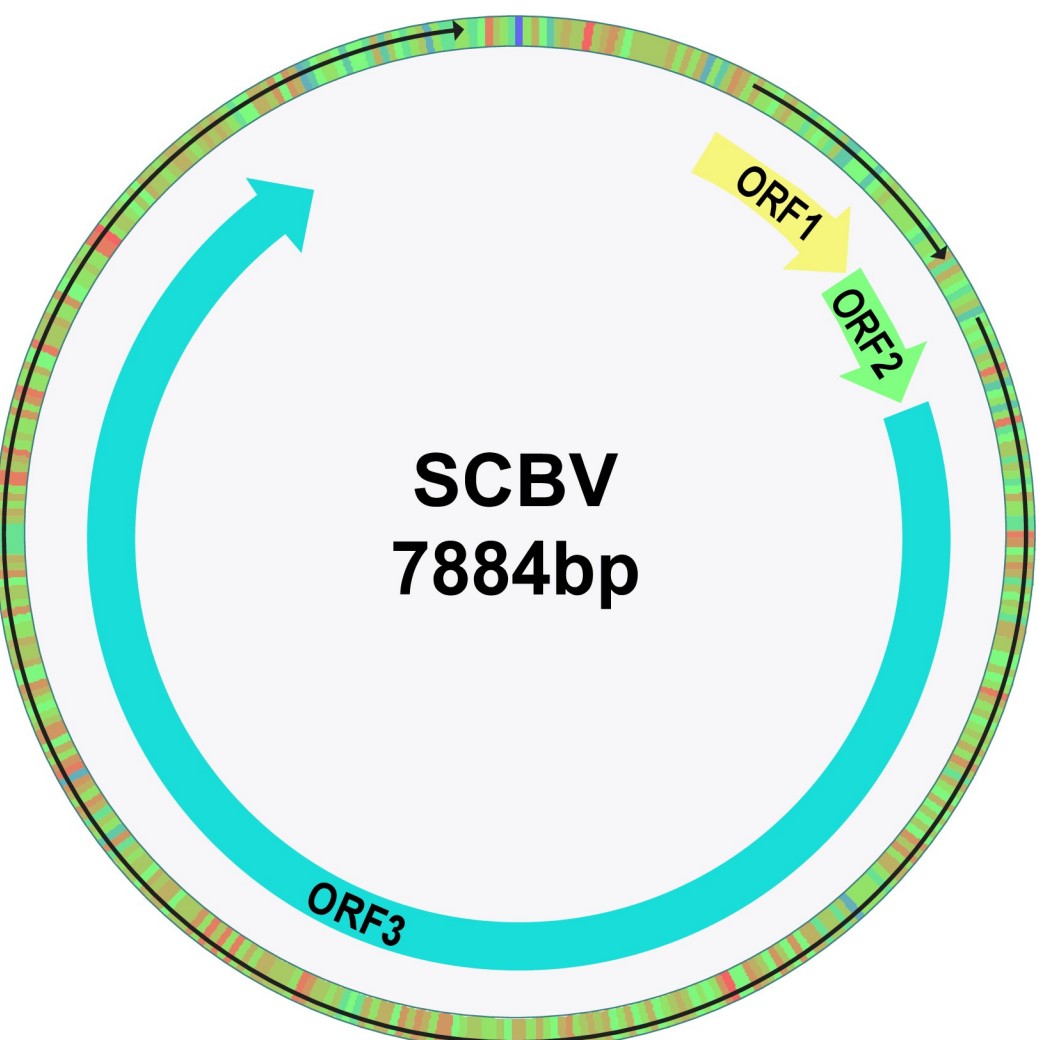

**Fig 2. Genomic organization of the sugarcane bacilliform virus.** The predicted ORFs denoted with arrows are composed of dsDNA that is 7884 bp in size.

### 3.3. ORF2 encoding DNA binding protein

A nucleic acid (DNA)-binding protein of the SCBV genome is encoded by ORF2 [6, 45]. RNA-hybrid and miRanda predicted potential target binding site of ssp-miR166 at locus position 1449–1470 (Fig 3A and 3C). Suitable candidate miRNAs from sugarcane (ssp-miR444 (a, b, 3p) were observed to target ORF2 at a single loci nucleotide position (1301–1326) as determined by the miRanda algorithm (Fig 3A). No sugarcane miRNAs were predicted to target the ORF2 gene with the RNA22 tool (Fig 3B). Similarly, RNAhybrid predicted the binding of ssp-miR166 at locus 1450 (Fig 3C). The miRNA prediction results revealed that no candidate miRNA was identified to have a potential genome binding site in the ORF2 region, as predicted by psRNATarget (Fig 3D).

### 3.4. ORF3 encoding polyprotein (CP, AP, RT, and RNase H)

The poly proteins constitute the largest portion of the SCBV genome encoded by ORF3 [2, 6]. Potential candidate miRNAs from sugarcane were identified by the miRanda algorithm to

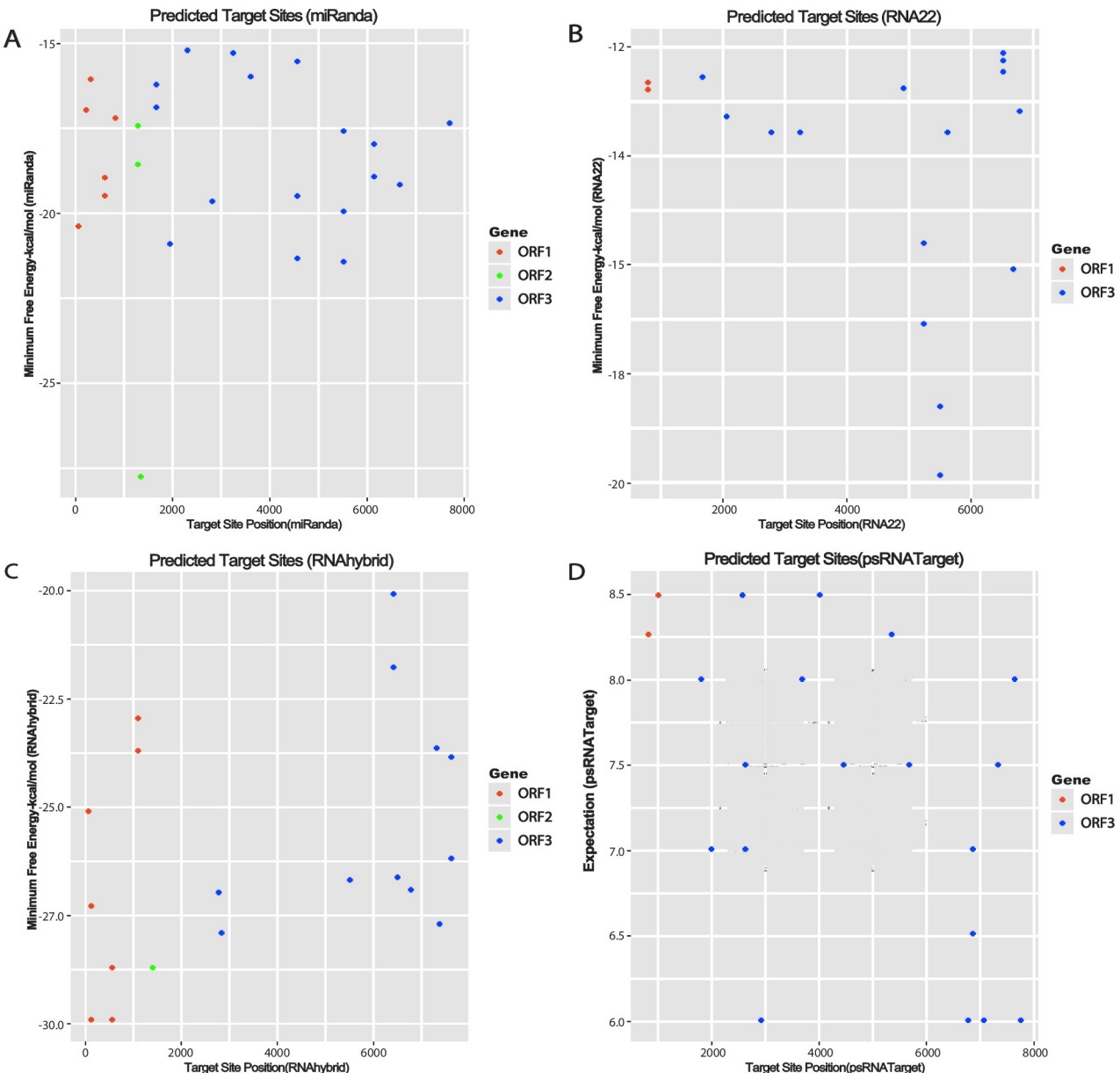

**Fig 3. Target prediction of sugarcane miRNAs in the SCBV genome.** Computational prediction of candidate miRNA targets in the genome of the SCBV. (A) miRNA targets obtained from miRanda. (B) RNA22 predicted potential hybridization sites. (C) Target sites of sugarcane miRNAs as identified by RNAhybrid. (D) Prediction results of target sites of sugarcane miRNAs as obtained by psRNATarget.

target ORF3, including sof-miR159 (a, b, c, d, and e) at common locus 5534, sof-miR167 (a, b) at locus 2273, sof-miR168b at locus 4588, sof-miR408 (a, b, c, d, and e) at the two common loci of 4595 and 6695, ssp-miR166 at locus 1986, ssp-miR827 at locus 2816, and ssp-miR444 (a, b, and c-3p) at common locus 6184. Multiple loci interactions were predicted for the sof-miR159, sof-miR408, and ssp-miR444 families at nucleotide positions (5534–5552, 5576–5596), (4595–4615, 6695–6715) and (1679–1701, 3293–3313) of ORF3, respectively (Fig 3A).

Potential target binding sites were determined for ORF3 of the SCBV genome by the RNA22 algorithm. These included sof-miR168a at locus 3263, sof-miR168b at nucleotide positions 1693 and 3263, sof-miR396 at locus 2050, and ssp-827 at locus 2796 (Fig 3B). Multiple

loci interactions were also identified for the sof-miR159, sof-miR408, and ssp-miR444 families at nucleotide positions (5532, 6536), (5645, 6695), and (5246, 6793) respectively (Fig 3B). Suitable miRNAs that potentially targeted ORF3 were hybridized in order to understand the miRNA—mRNA interaction via RNAhybrid. As a result of, sof-miR159 (a, b, d and e) was detected at common locus 5535, along with sof-miR159c at locus 6518, sof-miR167 (a, b) at locus 2826, sof-miR169 at locus 7362, ssp-miR473 (a, b, c) at common locus 6438, ssp-miR444 (a, b) at locus 6796, ssp-miR444 c-3p at locus 2899 and ssp-miR1432 at locus 7314 (Fig 3C). ORF3 was targeted by several candidate miRNAs, includingsof-miR159e at locus 2647, sof-miR396 at locus 5363, ssp-miR166 at locus 1986, ssp-miR437 (a, c) at locus 2647, ssp-miR827 at locus 7337, and ssp-miR444 (a, b and c-3p) at locus 6797, as identified by psRNATarget. Multiple loci interactions were observed for the sof-miR408 and ssp-miR444 families at the nucleotide positions of (1766–1786, 3669–3689, 5683–5702) and (4466–4486, 6797–6816, 6865–6885, 7079–7099), respectively (Fig 3D). The union plot indicates entire genome binding sites identified by the candidate miRNAs using target prediction tools (Fig 4, and S2 Table in S1 File).

## 3.5. Visualization and analysis of miRNA-target interaction network

Initially, the Circos plotting tool was designed to analyze mutations with comparative metagenomics and transcriptomic biological data [46]. To study a comprehensive visualization of host–virus interaction, we created a Circos plot to integrate biological data from sugarcane miRNAs and their predicted SCBV genomic target genes (ORFs) (Fig 5). In order to reduce visual graphical complexity and permit improved readability, we only used selected sugarcane miRNAs and their SCBV targets obtained from miRanda analysis. The miRanda algorithm considers seed-based interactions and the conservation level [47, 48]. The results suggest that biological data visualization of candidate miRNAs from sugarcane, with SCBV-encoded ORFs determines credible information of desirable preferred targets of SCBV ORFs using consensual miRNAs. We have combined sugarcane miRNA data and their predicted SCBV targets simultaneously in this manner.

## 3.6. Predicting common sugarcane miRNAs

Based on predicted targeting miRNAs from sugarcane to silence the SCBV genome, fourteen miRNAs (sof-miR156, sof-miR159c, sof-miR159e, sof-miR168a, sof-miR396, sof-miR408a, sof-miR408b, sof-miR408c, sof-miR408d, sof-miR408e, ssp-miR827, ssp-miR444a, ssp-miR444b and sof-miR444c-3p) were detected by union of consensus between the multiple algorithms (miRanda, RNA22, RNAhybrid and psRNATarget) used in this study (Fig 6). Moreover, SCBV genomic components (ORF1, ORF2, ORF3, and the large intergenic region (LIR)) were observed to be targeted by a total of eleven sugarcane miRNAs which were hybridized at unique positions within ORF1(sof-miR156 (locus 818) and sof-miR168 (a, b) (locus 617)) ORF2 (ssp-miR166 (locus 1450), ORF3 (sof-miR159c (locus 5534) and sof-miR408 (a, b, c, d and e) (locus 6695), and the LIR sof-miR396 (locus 79)) according to intersection between two consensual algorithms (Table 2, and S3 Table in S1 File).

## 3.7. Predicting consensual sugarcane miRNAs for silencing the SCBV genome

Out of 28 sugarcane miRNAs, only six sugarcane miRNA (sof-miR159 (a, b, d and e) at common locus position 5535 and ssp-miR444 (a, b) at locus 6797) were predicted at the common locus by at least three of the algorithms used (Fig 7 and Table 2). Out of 14 consensual miRNAs, only one miRNA of *S. officinarum* (sof-miR159e at locus 5535), with a MFE of -26.7

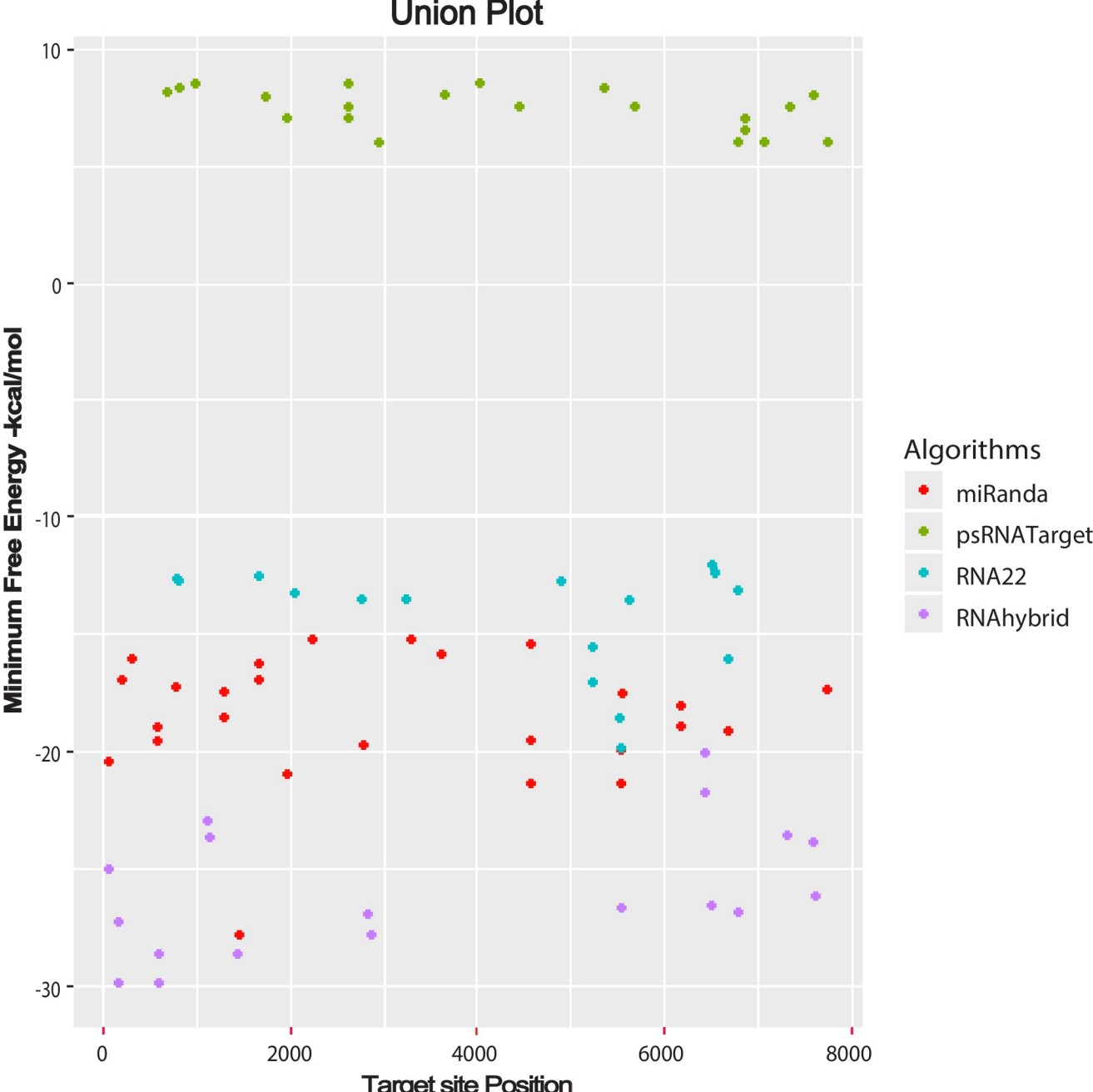

**Fig 4. Union plot representing all the predicted sugarcane miRNA targets in the SCBV genome.** miRNA target candidate prediction is represented as a union from all the algorithms used in this study.

Kcal/mol, was considered as the top effective candidate in terms of support more efficient silencing of the SCBV genome. The efficacy of the sof-miR159e target against SCBV was validated by the suppression of RNAi-mediated viral combat through the cleavage of viral mRNA or translational inhibition [43]. Multiple loci interactions were observed for sof-miR159e at nucleotide positions 5534–5552 (consensus of three algorithms, namely, miRanda, RNA22, and RNAhybrid) and 2647 (psRNATarget) of ORF3.

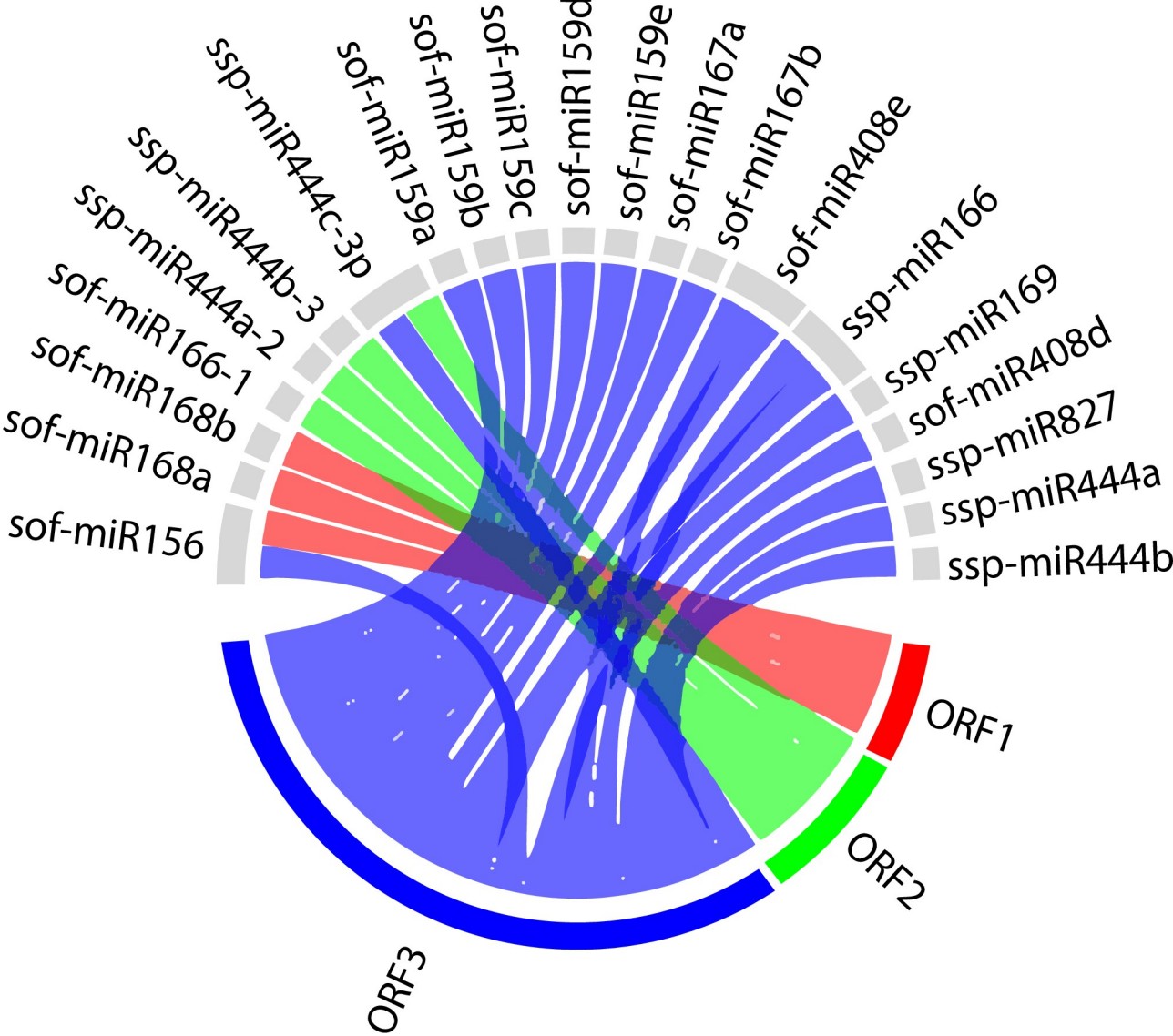

**Fig 5. Circos plot representing miRNA-target interaction.** Circos plot of genomic regulatory network interaction as predicted to be targeted by the sugarcane miRNAs. The red, green, and blue colored lines represent SCBV genome components (ORFs). The synergetic counterparts of sugarcane miRNAs and their target genes (ORFs) of the SCBV genome are interconnected with colored lines.

### 3.8. Prediction of consensus secondary structures

The validation of consensual sugarcane miRNAs was confirmed by the prediction of their stable secondary structures using the RNAfold algorithm. Precursors of mature sugarcane miRNAs were manually curated. The MFE is the key factor to determine the stable secondary structures of precursors. All the predicted consensual sugarcane miRNA precursors were observed to possess lower MFE values (ranging from −57.70 to −114.70 kcal/mol) (Table 3).

The predicted secondary structures of six precursors of pre-miRNAs are shown in (Fig 8), as predicted by the intersection of three consensual algorithms at the same locus. The top stable secondary structure of the sof-MIR159e precursor was predicted with standard features (MFE: 107.50 Kcal/mol, MFEI: 1.06 Kcal/mol). The predicted secondary structures of 14 consensual sugarcane miRNAs passed the aforementioned standard criteria. We have determined

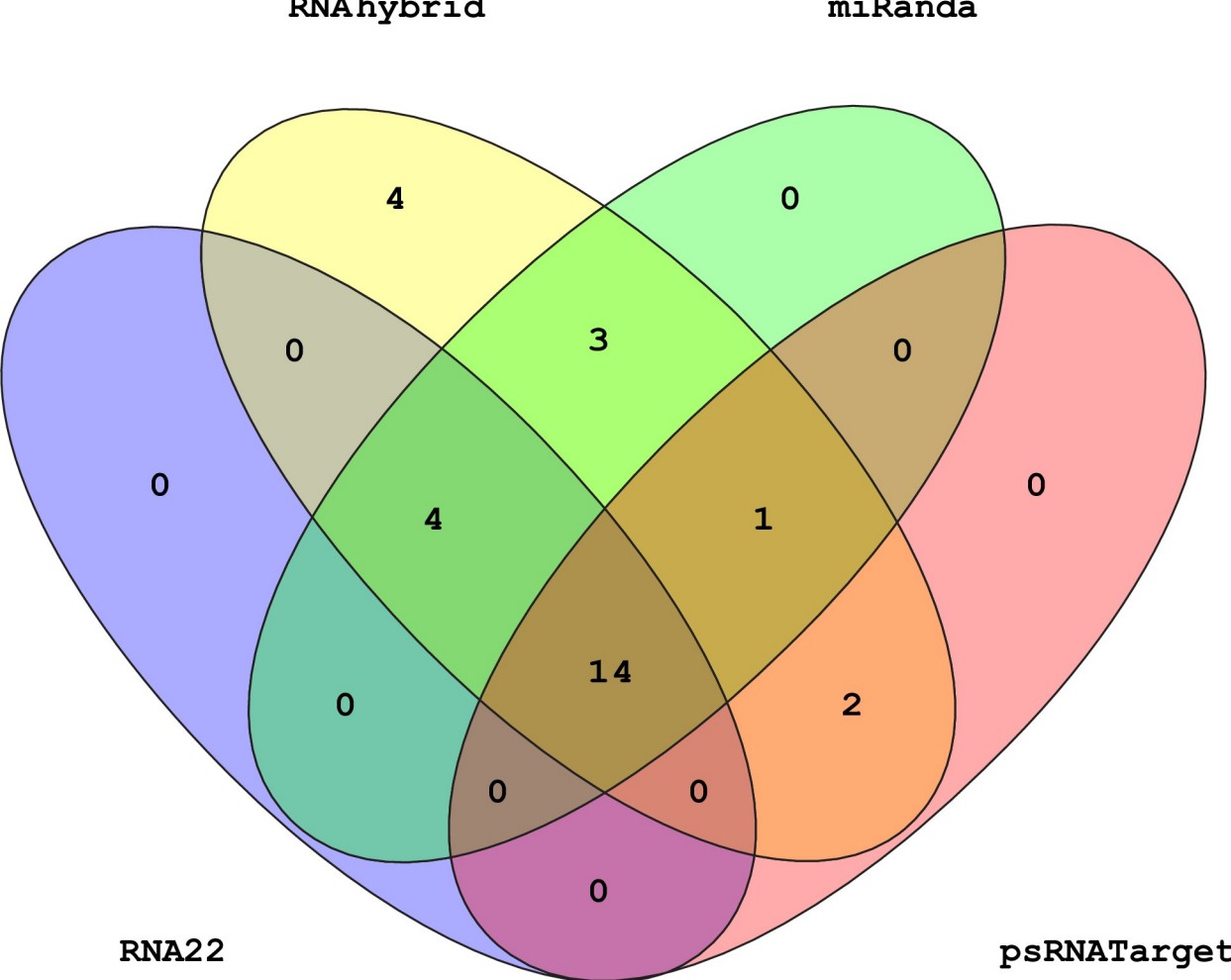

**Fig 6. Venn diagram plot of SCBV genome targeted by sugarcane miRNAs.** Venn diagram plot of the SCBV genome targeted by sugarcane miRNAs. In total, 28 loci are targeted by sugarcane miRNAs as predicted from four unique algorithms.

the salient characteristics of six consensus precursor miRNAs in this study, such as the MFE, AMFE, MFEI, length precursor, and GC contents. In our studies, the length precursor ranges from 105–266 nucleotides, along with a MFE of -57.70 to −110.70 kcal/mol, AMFE of -39.92 to 60.09, GC content of 38–47%, and MFEI from −0.83 to −1.26.

### 3.9. Assessment of free energy (ΔG) of miRNA-mRNA interaction

The predicted consensual sugarcane miRNAs were validated by estimating the free energies of miRNA/target duplexes (Table 3). The free energies (ΔG) of six consensual sugarcane miRNAs were estimated as follows: sof-miR159 (a, b, d) (ΔG: -20.10 kcal/mol), sof-miR15e (ΔG: -20.40 kcal/mol), and ssp-miR444 (a, b) (ΔG: -14.50 kcal/mol).

### 3.10. Tissue preferential expression analysis of sugarcane miRNAs

We used the "PmiRExAt" database to search for the expression analysis of the predicted sugarcane miRNAs. Homologous miRNAs were present in all three plant species, i.e., maize, rice, and wheat (S1–S3 Figs). The expression of these microRNAs was identified in all tissue types

**Table 2. Sugarcane miRNAs and their target positions in SCBV as identified by various algorithms.**

| Sugarcane miRNAs | Position miRanda | Position RNA22 | Position RNAhybrid | Position psRNATarget | MFE* miRanda | MFE** RNA22 | MFE RNAhybrid | Expectation psRNATarget |
|---|---|---|---|---|---|---|---|---|
| sof-miR156 | 818 | 817 | 7608 | 7609 | -17.23 | -12.7 | -23.9 | 8 |
| sof-miR159a | 5534 | 5532 | 5535 | | -21.45 | -19.9 | -26.7 | |
| sof-miR159a(1) | 5576 | 6536 | | | -17.54 | -12.5 | | |
| sof-miR159b | 5534 | 5532 | 5535 | | -21.45 | -19.9 | -26.7 | |
| sof-miR159b(1) | 5576 | 6536 | | | -17.54 | -12.5 | | |
| sof-miR159c | 5534 | 5532 | 6518 | 1003 | -20.02 | | -28 | 6 |
| sof-miR159c(1) | | 6533 | | | | -12.1 | | |
| sof-miR159d | **5534** | **5532** | **5535** | | -21.45 | -19.9 | -26.7 | |
| sof-miR159d(1) | 5576 | 6536 | | | -17.54 | -12.5 | | |
| sof-miR159e | 5534 | 5532 | 5535 | | -21.45 | -19.9 | -26.7 | |
| sof-miR159e(1) | 3633 | 6536 | | | -16 | -12.1 | | |
| sof-miR167a | 2273 | | 2826 | | -15.24 | | -27 | |
| sof-miR167b | 2273 | | 2826 | | -15.24 | | -27 | |
| sof-miR168a | 617 | 834 | 612 | 4046 | -19.53 | -12.8 | -29.9 | 8.5 |
| sof-miR168a(1) | | 3263 | | | | -13.6 | | |
| sof-miR168b | 617 | 1693 | 612 | | -19 | -26.7 | -28.7 | |
| sof-miR168b(1) | 4588 | 4907 | | | -15.49 | -12.8 | | |
| **sof**-miR396 | 79 | 2050 | 79 | 5563 | -20.44 | -13.3 | -25.1 | 8.25 |
| **sof**-miR408a | 6695 | 6695 | 174 | 3669 | -19.19 | -16.1 | -27.3 | 8 |
| sof-miR408a(1) | 4595 | 5645 | | 1766 | -21.35 | -13.6 | | 8 |
| **sof**-miR408b | 6695 | 6695 | 174 | 3669 | -19.19 | -16.1 | -27.3 | 8 |
| sof-miR408b(1) | 4595 | 5645 | | 1766 | -21.35 | -13.6 | | 8 |
| **sof**-miR408c | 6695 | 6695 | 174 | 3669 | -19.19 | -16.1 | -27.3 | 8 |
| sof-miR408c(1) | 4595 | 5645 | | 1766 | -21.35 | -13.6 | | 8 |
| **sof**-miR408d | 6695 | 6695 | 174 | 3669 | -19.19 | -16.1 | -27.3 | 8 |
| sof-miR408d(1) | 4595 | 5645 | | 1766 | -21.35 | -13.6 | | 8 |
| **sof**-miR408e | 6695 | 6695 | 174 | 5683 | -19.19 | -16.1 | -29.9 | 7.5 |
| sof-miR408e (1) | 4595 | 5645 | | 1766 | -19.53 | -13.6 | | 8 |
| sof-miR408e (2) | 242 | | | | -17.01 | | | |
| ssp-miR166 | 1449 | | 1450 | 7750 | -27.85 | | -28.7 | 6 |
| ssp-miR166(1) | 1986 | | | 1986 | -20.95 | | | 7 |
| ssp-miR169 | 7748 | | 7362 | | -17.38 | | -27.7 | |
| ssp-miR437a | | | 6438 | 2646 | | | -20.1 | 7 |
| ssp-miR437b | | | 6438 | | | | -20.1 | |
| ssp-miR437c | | | 6437 | 2647 | | | -21.8 | 7.5 |
| ssp-miR437c(1) | | | | 2974 | | | | 6 |
| ssp-miR528 | | | 7619 | | | | -26.2 | |
| ssp-miR827 | 2816 | 2796 | 1170 | 7337 | -19.73 | -13.6 | -23.7 | 7.5 |
| ssp-miR444a | 6184 | 6793 | 6796 | 6797 | -18.04 | -13.2 | -26.9 | 6 |
| ssp-miR444a(1) | 3293 | 5246 | | 6865 | -15.23 | -15.6 | | 6.5 |
| ssp-miR444a(2) | 1301 | | | | -17.44 | | | |
| ssp-miR444b | 6184 | 6793 | 6796 | 6797 | -18.04 | -13.2 | -26.9 | 6 |
| ssp-miR444b(1) | 3293 | 5246 | | 7079 | -15.23 | -15.6 | | 6 |
| ssp-miR444b(2) | 1301 | | | 4466 | -17.44 | | | 7.5 |
| ssp-miR444b(3) | 1676 | | | | -16.25 | | | |
| ssp-miR444c-3p | 6184 | 5246 | 2899 | 6797 | -18.95 | -17.1 | -27.9 | 6 |

(*Continued*)

**Table 2.** (Continued)

| Sugarcane miRNAs | Position miRanda | Position RNA22 | Position RNAhybrid | Position psRNATarget | MFE* miRanda | MFE** RNA22 | MFE RNAhybrid | Expectation psRNATarget |
|---|---|---|---|---|---|---|---|---|
| ssp-miR444c-3p (1) | 328 | | | 6865 | | -16.8 | | 7 |
| ssp-miR444c-3p (2) | 1301 | | | | | -18.59 | | |
| ssp-miR444c-3p (3) | 1680 | | | | | -16.96 | | |
| ssp-miR1128 | | | 1137 | | | | -23 | |
| ssp-miR1432 | | | 7314 | | | | -23.6 | |

*MFE: Minimum free energy measured in /Kcal/mol where *MFE represents minimum folding energy measured in Kcal/mol.

in each species. Therefore, the expression of sugarcane miRNAs was confirmed in other plant species, i.e., maize, rice, and wheat. Evidence of the existence of the same miRNAs in sugarcane is also provided. Most of the stated miRNAs have also been confirmed, in multiple studies, for their expression and roles in plant cellular pathways [49, 50].

## 4. Discussion

For the filtering of false positive results, we studied the effectiveness of the computational algorithms considered here to validate the miRNA target prediction data. We designed an effective approach for the validation of miRNA target prediction results at individual, union, and intersection levels. Computational prediction algorithms offer rapid methods to identify potential host-derived miRNA targets in virus genomes. Default parameters represent optimized specifications for each miRNA to its respective target site in the viral genome. This varies with respect to each algorithm/tool and can be modified for fine-tuning the settings or increasing the level of sensitivity for predicted sites. Default parameters are effective for screening out false-positive attachment sites for miRNAs using multiple prediction tools. miRanda is a widely used algorithm that includes the main aspects of miRNA–target prediction, such as the conservation level and miRNA 3'UTR site [51]. The RNA22 algorithm is a novel alternative option for exploring new miRNA–mRNA interactions because of its unique capabilities—although it has a high likelihood of generating false-positive results [47]. We calculated the MFE and determined the target inhibition as recommended by Broderson by using RNAhybrid [37].

Several potential sugarcane miRNA targets and miRNA–mRNA interactions could be consensually predicted by all of the algorithms (Fig 7). Plant miRNAs are responsible for inducing the degradation of the target genes using perfect or imperfect complementarity base pairing [52]. The current study demonstrates that SCBV genome components (ORF1, ORF2, and ORF3) are susceptible to targeting by a set of consensual sugarcane miRNAs. In addition, sof-miR159 (a, b, d, and e) was found to target ORF3 at a consensual hybridization site by at least three algorithms (Fig 8). Free energy assessment is a dynamic feature of miRNA and target binding. Previous studies have revealed a significant correlation of free energy between the translational repression and the hybridization binding of the seed region [53]. The thermodynamic stability of the miRNA–mRNA duplex was estimated by the assessment of free energy to monitor site accessibility for the determination of the secondary structure duplex [27]. In order to validate miRNA–mRNA interaction, the free energy of a duplex was assessed (Table 2). Our prediction results show high stability for the sugarcane-encoded miRNA–

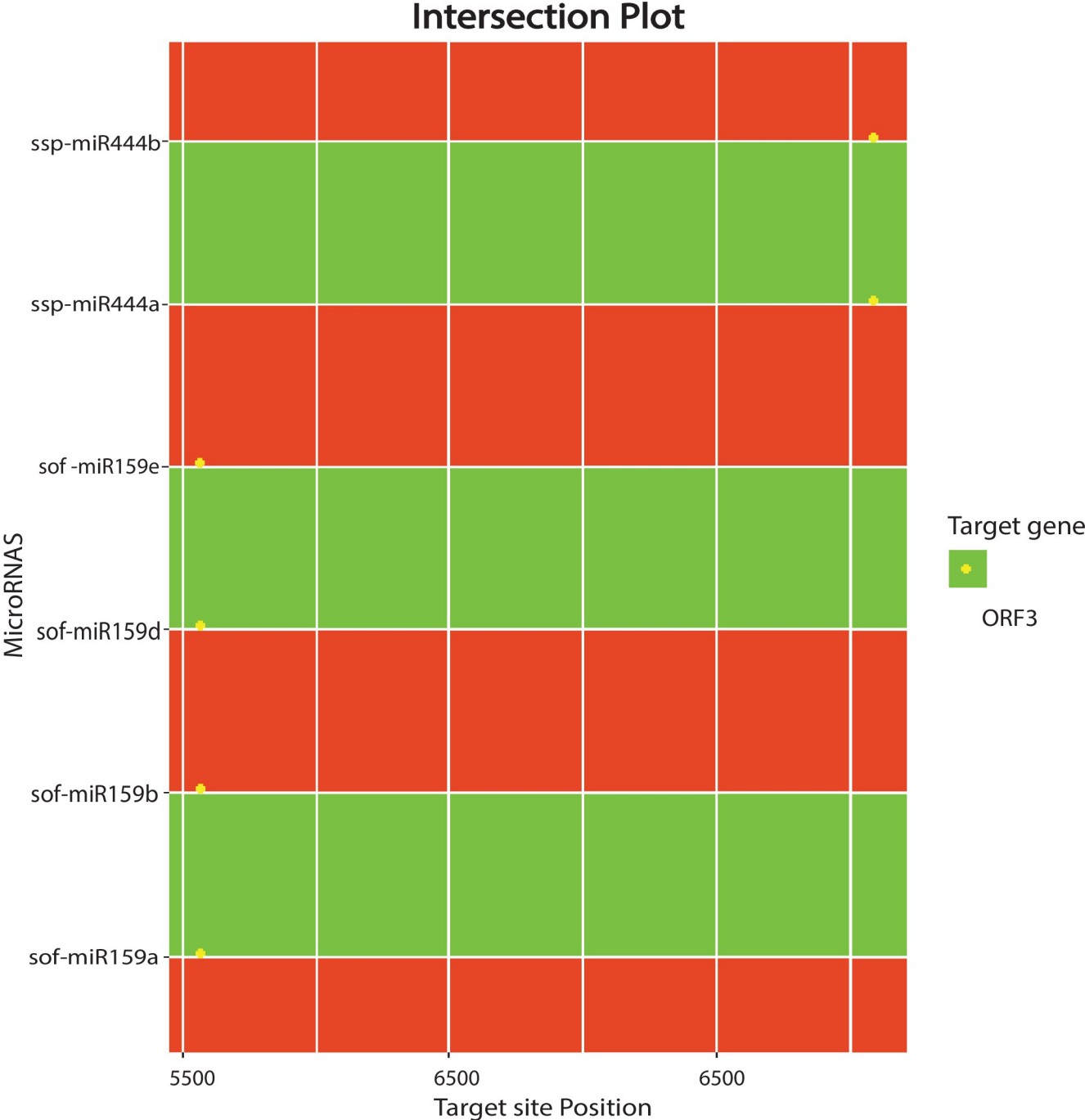

**Fig 7. Intersection plot of sugarcane miRNAs predicted from at least three algorithms.** The intersection plot was created with the miRNAs predicted from at least three algorithms (miRanda, RNA22 and RNAhybrid). Color codes given within the figure.

SCBV-mRNA duplex at a low free energy level (Table 3 and Fig 8). The RNA duplex is considered to be more stable due to the stronger binding of miRNA to mRNA [54, 55].

We used union and intersection approaches to reduce false positive prediction. Union approaches rely on combining more than one target prediction tool when finding true and false targets. The sensitivity level for a predicted target increases due to a decrease in specificity.

**Table 3. The salient parameters of precursor miRNAs were determined along with the estimation of free energy.**

| miRNA ID | Length miRNA | Length precursor | MFE[1] (Kcal/mol) | AMFE[2] | MFEI[3] | (G+C)% | ΔG[4] (Kcal/mol) |
|---|---|---|---|---|---|---|---|
| sof-miR156 | 20 | 137 | -66.20 | -48.32 | -0.96 | 50.00 | -14.30 |
| sof-miR159a | 21 | 265 | -110.30 | -41.62 | -0.87 | 47.60 | -20.10 |
| sof-miR159b | 21 | 266 | -110.30 | -41.46 | -0.87 | 47.60 | -20.10 |
| sof-miR159c | 21 | 238 | -110.60 | -46.47 | -0.88 | 52.38 | -19.70 |
| sof-miR159d | 21 | 265 | -105.80 | -39.92 | -0.83 | 47.60 | -20.10 |
| sof-miR159e | 21 | 264 | -107.50 | -40.71 | -1.06 | 38.09 | -20.40 |
| sof-miR168a | 21 | 104 | -66.20 | -63.65 | -1.02 | 61.90 | -18.20 |
| sof-miR396 | 21 | 134 | -67.40 | -50.29 | -1.17 | 42.85 | -19.60 |
| sof-miR408a | 21 | 283 | -114.70 | -40.53 | -0.60 | 66.66 | -16.00 |
| sof-miR408b | 21 | 286 | -113.20 | -39.58 | -0.59 | 66.66 | -16.00 |
| sof-miR408c | 21 | 286 | -115.80 | -40.48 | -0.60 | 66.66 | -16.00 |
| sof-miR408d | 21 | 215 | -79.00 | -36.76 | -0.55 | 66.66 | -16.00 |
| sof-miR408e | 21 | 283 | -99.00 | -34.98 | -0.56 | 61.90 | -16.00 |
| ssp-miR827 | 21 | 130 | -64.00 | -49.23 | -1.29 | 38.09 | -17.90 |
| ssp-miR444a | 21 | 105 | -57.70 | -54.95 | -1.15 | 47.62 | -14.50 |
| ssp-miR444b | 21 | 106 | -63.70 | -60.09 | -1.26 | 47.62 | -14.50 |
| ssp-miR444c | 21 | 108 | -61.80 | -57.22 | -1.33 | 42.85 | -15.30 |

[1] MFE is minimum free energy.

[2] AMFE represents adjusted minimum free energy.

[3] MFEI defines as minimum free energy index.

[4] ΔG represents minimum free energy of duplex formation.

An intersection approach is entirely different and depends upon the combination of two or more computational tools and enhances the specificity level of predicted targets due to a decrease in sensitivity [56]. Our target prediction results revealed that both computational approaches achieved the best outcomes with maximum performance for predicting and estimating the best targets (Figs 6 and 7). Previous studies have also reported the silencing of plant viruses using host-derived miRNAs when applying a set of computational algorithms. The identification and evaluation of best-fit candidate miRNA targets for different plants has been concluded successfully with potato virus Y (PVY) [57], maize chlorotic mottle virus (MCMV) [58], CLCuKoV-Bu [59], rice yellow mottle virus (RYMV), [60] and SCBGAV to find miRNA–target interaction [61]. We have designed an equal novel bioinformatics approach for target prediction in the SCBV genome to control the emerging presence of *Badnavirus* in sugarcane cultivars.

In our previous study, we identified the most ideal consensual sugarcane miRNA (sof-miR396) to target ORF3 of the SCBGAV genome using multiple computational algorithms [61]. The quantity of false positive miRNA–target interaction estimated by multiple algorithms depends upon the mode of miRNA–target recognition. MFE is also another important factor that affects miRNA–target interaction in result validation [62]. To set a lower MFE value will give rise to a higher probability of miRNA–target complex formation [63]. In the current study, for miRanda analysis, a stringent cut-off point of −15 kcal/mol was set for narrowing down the miRNA candidates. Similarly, to validate host–virus interaction, a MFE cut-off point of -20 kcal/mol applied for RNAhybrid analysis [32].

Although MFE has a considerable role for development of miRNA–mRNA complexes, it does not certify that interactions will lead to functional changes. In the current study, we have identified six potential miRNA hybridization binding sites that have exhibited low MFEs and free energy for duplex formation. These predicted miRNAs not only have potential targets for

sof-MIR159a    sof-MIR159b    sof-MIR159d    sof-MIR159e    ssp-MIR444a    ssp-MIR444b

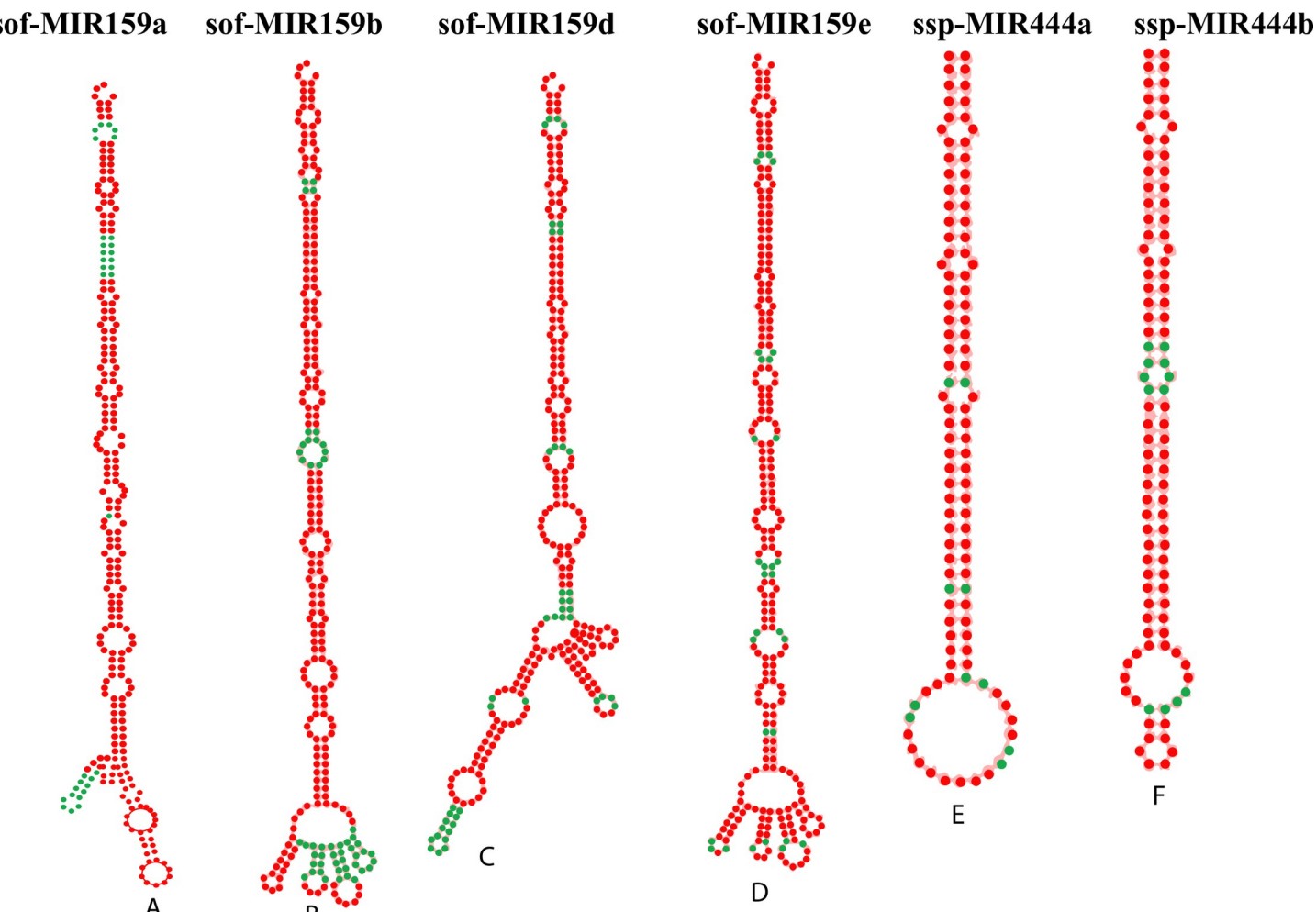

**Fig 8. Prediction of secondary structures of stem-loop sequences of sugarcane miRNAs.** Six pre-miRNA secondary structures (precursors of sugarcane miRNAs) were identified in this study by consensus between three algorithms. The sugarcane mature miRNA name IDs, accession IDs, MFEs and MFEIs are given as follows: (A) sof-MIR159a (MI0001756), -110.30 kcal/mol, -0.87 B) sof-MIR159b (MI0001757), -110.30 kcal/mol, -0.87; (C) sof-MIR159d (MI0001758), -105.80 kcal/mol, -0.83; (D) sof-MIR159e (MI0001759), -107.50 kcal/mol, -1.06; (E) ssp-MIR444a (MI0018185), -57.70 kcal/mol, -1.15; (F) ssp-MIR444b (MI0018186), -63.70kcal/mol, -1.26.

the SCBV genome at the transgenic level but also have a stronger probability to develop miRNA–viral mRNA complex formation. These miRNAs also have chance to participate in a SCBV replication mechanism, where a consensus sugarcane miRNA (sof-miR396) has a binding site within the SCBV large intergenic region (LIR) at locus 79 as predicted by the miRanda and RNAhybrid algorithms. In the previous study, we predicted that sof-miRNA396 is an effective candidate to target the SCBGAV genome [61]. Notably, sof-miR159e was predicted by all the algorithms. Additionally, miR159 was explored and was found to present a strong role for silencing *GAMYB* to enable normal growth [64]. Phe-MIR159 involved in regulating the gene responsible for secondary thickening in *Phyllostachys edulis* [65]. It is important to assess the function of predicted potential consensual miRNAs for the identification of *Badnavirus* replication to demonstrate SCBV replication experimentally. A hypothetical model was designed to show that sugarcane-derived miRNAs can inhibit SCBV mRNA and sugarcane genes against SCBV virus (Fig 9). It facilitates plant-encoded miRNAs in the cleavage of SCBV miRNA.

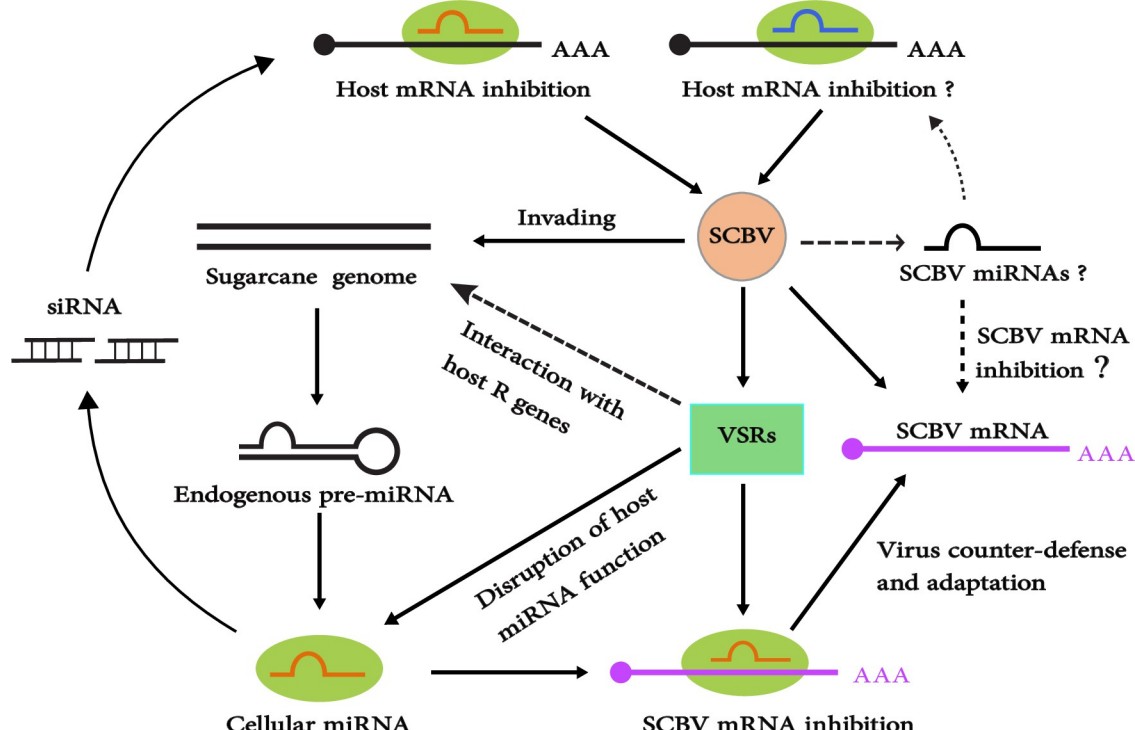

**Fig 9. Schematic model designed for miRNA-mediated gene silencing in plant-virus interaction.** SCBV can activate the production of endogenous sugarcane miRNAs post infection. Moreover, sugarcane miRNAs can target SCBV mRNA for degradation.

RNAi screening is a novel technology for discovering various cellular functions and identifying host-derived factors of viruses [66]. Here, we selected 28 experimentally validated sugarcane miRNAs with annotated targets that are part of SCBV. amiRNA-based silencing technology has been successfully validated in many crop plants for controlling emerging plant viruses [23, 24, 26]. In summary, our computational work for SCBV genome silencing could offer a new approach for the production of antiviral agents. Furthermore, we demonstrated a method to minimize the novel antiviral effects of host-derived miRNAs against SCBV.

## 5. Conclusions

SCBV has appeared as a major problem in China. SCBV diminishes quantitative yields in all sugarcane cultivars. In the current study, prior to cloning, we have applied computational tools to predict and comprehensively analzse candidate miRNA from sugarcane against SCBV. Among them, sof-miR159e was predicted as the top effective candidate that could target the vital gene (ORF3) of the SCBV genome. Our results demonstrate an alternative strategy to existing molecular approaches that could be repurposed to control badnaviral infections. The current findings provide *in silico* evidence of a novel scheme to construct miRNA-mediated gene silencing therapeutics to combat SCBV.

## Supporting information

**S1 File.**
(RAR)

**S1 Fig. Tissue preferential expression heatmap of sugarcane miRNAs in maize.**
(TIF)

**S2 Fig. Tissue preferential expression heatmap of sugarcane miRNAs in rice.**
(TIF)

**S3 Fig. Tissue preferential expression heatmap of sugarcane miRNAs in wheat.**
(TIF)

## Acknowledgments

We are highly thankful to our lab colleagues for their assistance in data analysis. We wish to thank Dr. Zhiqiang Xia (ITBB) for providing facilities and assistance in the construction of the Circos plot.

## Author Contributions

**Conceptualization:** Muhammad Aleem Ashraf, Shuzhen Zhang.

**Data curation:** Muhammad Aleem Ashraf, Xiaoyan Feng, Linbo Shen, Shuzhen Zhang.

**Formal analysis:** Muhammad Aleem Ashraf, Xiaowen Hu.

**Funding acquisition:** Xiaoyan Feng, Xiaowen Hu, Shuzhen Zhang.

**Investigation:** Muhammad Aleem Ashraf, Xiaoyan Feng, Linbo Shen.

**Methodology:** Muhammad Aleem Ashraf, Xiaowen Hu, Fakiha Ashraf, Muhammad Shahzad Iqbal.

**Project administration:** Shuzhen Zhang.

**Resources:** Xiaoyan Feng.

**Software:** Muhammad Aleem Ashraf, Xiaowen Hu, Fakiha Ashraf, Muhammad Shahzad Iqbal.

**Supervision:** Shuzhen Zhang.

**Validation:** Muhammad Aleem Ashraf, Xiaoyan Feng, Linbo Shen.

**Writing – original draft:** Muhammad Aleem Ashraf, Fakiha Ashraf, Shuzhen Zhang.

**Writing – review & editing:** Muhammad Aleem Ashraf, Fakiha Ashraf, Shuzhen Zhang.

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
