## [Decision Letter · Decision Letter 0]

15 Jan 2021

PONE-D-20-35060

An Algorithmic framework for genome-wide identification of Sugarcane (Saccharum officinarum L.)-encoded microRNA targets against SCBV

PLOS ONE

Dear Dr. Asharf,

Thank you for submitting your manuscript to PLOS ONE. After careful consideration, we feel that it has merit but does not fully meet PLOS ONE’s publication criteria as it currently stands. Therefore, we invite you to submit a revised version of the manuscript that addresses the points raised during the review process.

We look forward to receiving your revised manuscript.

Kind regards,

Edwin Wang

Academic Editor

PLOS ONE

Journal Requirements:

2. Please amend your list of authors on the manuscript to ensure that each author is linked to an affiliation. Authors’ affiliations should reflect the institution where the work was done (if authors moved subsequently, you can also list the new affiliation stating “current affiliation:….” as necessary).

Reviewers' comments:

Reviewer's Responses to Questions

**Comments to the Author**

1. Is the manuscript technically sound, and do the data support the conclusions?

Reviewer #1: Partly

Reviewer #2: Partly

2. Has the statistical analysis been performed appropriately and rigorously? 

Reviewer #1: Yes

Reviewer #2: Yes

3. Have the authors made all data underlying the findings in their manuscript fully available?

Reviewer #1: Yes

Reviewer #2: Yes

4. Is the manuscript presented in an intelligible fashion and written in standard English?

Reviewer #1: Yes

Reviewer #2: Yes

5. Review Comments to the Author

Reviewer #1: Ashraf et al., have attempted to contribute in addressing the situation caused by SCBV to sugarcanes by envisaging miRNAs that could be helpful in the hushing of SCBV. This study sounds quite useful to me in terms of the economical burden and plants’ health; however, I have come across quite a few major and minor concerns that must be addressed before the manuscript could be accepted for publication.

Major concerns

(1) The title of the manuscript doesn’t complement the content. It gives a feeling of some novel algorithmic framework design; however, the study doesn’t directly design or improve any algorithm but just entirely depends on the existing platforms. Please modify the title to avoid misguidance.

(2) There is obstruction in the flow of the Abstract portion, please rearrange the sentences and remove any redundant detail.

(2) Introduction part, the authors mention names like RNAi technology and amiRNA approach, however, I didn’t find enough detail about these approaches in the text. They should be explained for the reader with citations.

(3) Line 135 (Page-7), why the authors maintained the default parameters? What makes them sure that the default parameters are suitable in this particular case, because though the default settings are universally applied in most of the conditions, these algorithms could get tricky in situations and they need to be tuned for a particular case. Is there any citation?

(4) Line 138 (Page-7) RNA22, what does the author mean by highly sensitive? This word doesn’t make sense to me. What’s the difference between a sensitive and an insensitive server?

(5) Please provide the citations for the all the maintained or adjusted parameters in case of every server used.

(6) In case of mutated viruses, do the outcomes of this study remains the same? The authors should consider variations in the genome since they could greatly alter the genomic behavior.

(7) Why the algorithms fail to compute data in most of the cases as can be seen in Table 2? Did the authors try to tune the parameters for computing the data?

(8) Discussion portion line 1, authors have mentioned that the performance of the computational algorithms is studied which is contradictory to the methodological section where the algorithms are only used for computing the data, not for evaluating or setting any benchmarks for comparisons. Please correct your statement, and search for similar confusions in the manuscript.

(9) What are the future prospects of this study? How the results could be considered further in terms of the computational drug design.

(10) Please read the manuscript by a native or at least a proficient English user, as I have encountered multiple errors in the punctuation, indentation, and word choice. I have highlighted few of them in the minor concerns but look for other similar issue thoroughly.

Minor concerns

(a) Line 16 (P-1), expand the full form of ORFs.

(b) Line 28 (P-1), “Consequently……mutants faster”, the sentence is grammatically wrong.

(c) Line 42 (P-2), avoid extra spacing.

(d) Line 42 (P-2), that was composed OR that is composed?

(e) Line 61 (P-3), Plant employs, the word employ is not suitable here as plants use their internal mechanism of defense, instead change it to “possess” or “enriched with” …

(f) Citation for pDRAW32 is missing.

(e) Line 104 (Page-5) The sentence has unnecessary detail and causes confusion. Instead, just write that the transcript is published and available via accession no. X, and provide the citation. Words like cloning, submission to NCBI etc. are not needed.

(g) Line 180 (Page-9), R is not a language, it’s a statistical environment. Please correct.

(h) Discussion section, no detail is given about the future prospects of this work

(i) Language needs to be highly improved.

Reviewer #2: In this study, the authors predicted and comprehensively evaluated sugarcane miRNAs for the silencing of SCBV genome using in-silico algorithms. They used several tools to predict the miRNA targets and identified several potential candidate miRNAs which may be used for the silencing of SCBV. However, the authors did not provide the expression of these miRNAs and predict their targets in the host sugarcane genome, which should be considered when selecting siRNA.

6. PLOS authors have the option to publish the peer review history of their article (what does this mean?). If published, this will include your full peer review and any attached files.

Reviewer #1: No

Reviewer #2: No

---

## [Author Response · Author response to Decision Letter 0]

23 Mar 2021

Response to Reviewer’s Comments

Dear Editor,

Coauthors and I very much appreciated the encouraging, critical and constructive comments on this manuscript by the reviewers. The comments have been very thorough and useful in improving the manuscript. We strongly believe that the comments and suggestions have increased the scientific value of revised manuscript by many folds. We have taken them fully into account in revision. We are submitting the corrected manuscript with the suggestion incorporated the manuscript. The manuscript has been revised as per the comments given by the reviewers, and our responses to all the comments are as follows: 

Reviewer #1: 

 Ashraf et al., have attempted to contribute in addressing the situation caused by SCBV to sugarcanes by envisaging miRNAs that could be helpful in the hushing of SCBV. This study sounds quite useful to me in terms of the economic burden and plants’ health; however, I have come across quite a few major and minor concerns that must be addressed before the manuscript could be accepted for publication.

(Major Concerns)

1. The title of the manuscript doesn’t complement the content. It gives a feeling of some novel algorithmic framework design; however, the study doesn’t directly design or improve any algorithm but just entirely depends on the existing platforms. Please modify the title to avoid misguidance.

.

Response- Thank you so much for your suggestion. As per reviewer suggestion, correction has been made in the revised manuscript and we have modified the title. We have removed words “An Algorithmic framework for” from the title and added “In silico”. Please see line 1.

2. There is obstruction in the flow of the Abstract portion, please rearrange the sentences and remove any redundant details. Introduction part, the authors mention names like RNAi technology and amiRNA approach, however, I didn’t find enough detail about these approaches in the text. They should be explained for the reader with citations.

Response- Thank you very much for your valuable suggestion. Correction has been made as per reviewer suggestion. Redundant details have been removed. Please see line 21-23. Correction has been made as per reviewer suggestion. The introduction section is greatly improved with new text. Please see 62-67 lines for RNAi and lines 78-82 for amiRNA.

3. Line 135 (Page-7), why the authors maintained the default parameters? What makes them sure that the default parameters are suitable in this particular case, because though the default settings are universally applied in most of the conditions, these algorithms could get tricky in situations and they need to be tuned for a particular case. Is there any citation?

Response- Thank you so much for your comment. As per reviewer suggestion, correction has been made in the revised manuscript. It was written wrongly. We have replaced it with ‘standard’ instead of ‘default’. Sorry it type error. Please see line 145 of the revised manuscript. 

4. Line 138 (Page-7) RNA22, what does the author mean by highly sensitive? This word doesn’t make sense to me. What’s the difference between a sensitive and an insensitive server?

Response- Thank you so much for your suggestion. RNA22 does not rely upon cross-species conservation, is resilient to noise, and, unlike previous methods, it first finds putative microRNA binding sites in the sequence of interest, then identifies the targeting microRNA. As per reviewer suggestion, ‘highly sensitive’ is removed. Please see line 148. 

5. Please provide the citations for all the maintained or adjusted parameters in case of every server used. 

Response- Thank you very much for your valuable suggestion. As per reviewer suggestion, correction has been made in the revised manuscript. We used the server as such.

6. In case of mutated viruses, do the outcomes of this study remain the same? The authors should consider variations in the genome since they could greatly alter the genomic behavior.

Response- Thank you very much for your valuable suggestion. As per reviewer suggestion, correction has been made in the revised manuscript. After mutation in SCBV viruses, the results will greatly change but it depends upon the kind of mutation in the sequences. 

7. Why the algorithms fail to compute data in most of the cases as can be seen in Table 2? Did the authors try to tune the parameters for computing the data?

Response- Thank you so much for your comment. We did not tune any parameters during computing the data. The results are as usual the same as predicted by the algorithms.

8. Discussion portion line 1, authors have mentioned that the performance of the computational algorithms is studied which is contradictory to the methodological section where the algorithms are only used for computing the data, not for evaluating or setting any benchmarks for comparisons. Please correct your statement, and search for similar confusions in the manuscript.

Response- Thank you so much for your comments. Correction has been made per reviewer suggestion. We have removed the word “performance” and placed it with ‘effectiveness’ Please see line 346.

9. What are the future prospects of this study? How the results could be considered further in terms of the computational drug design?

Response- Thank you so much for your comments. Correction has been made in the revised manuscript. We have already showed text in discussion for future work. Please see line 420-424, and 426-434 of the revised manuscript.

10. Please read the manuscript by a native or at least a proficient English user, as I have encountered multiple errors in the punctuation, indentation, and word choice. I have highlighted few of them in the minor concerns but look for other similar issue thoroughly?

Response- Thank you so much for your comment. As per reviewer suggestion, correction has been made in the revised manuscript. The manuscript has now been edited by acquiring English correction services from the Mdpi Experts.

(Minor Concerns)

a. Line 16 (P-1), expand the full form of ORFs. 

Response- Thank you so much for your comment. Correction has been made as per reviewer suggestion. Please see line 15.

b. Line 28 (P-1), “Consequently……mutants faster”, the sentence is grammatically wrong.

Response- Thank you so much for your comment. Correction has been made as per reviewer suggestion. Please see lines 26-27.

c. Line 42 (P-2), avoid extra spacing? 

Response- Thank you so much for your comments. Correction has been made as per reviewer suggestion.

d. Line 42 (P-2), that was composed OR that is composed?

Response- Thank you so much for your comment. Correction has been made as per reviewer suggestion. Please see line 41.

e. Line 61 (P-3), Plant employs, the word employ is not suitable here as plants use their internal mechanism of defense, instead change it to “possess” or “enriched with”. Line 104 (Page-5) the sentence has unnecessary detail and causes confusion. Instead, just write that the transcript is published and available via accession no. X, and provide the citation. Words like cloning, submission to NCBI etc. are not needed.

Response- Thank you so much for your comments. Correction has been made as per reviewer suggestion. Please see line 61 and 115.

f. Citation for pDRAW32 is missing.

Response- Thank you so much for your comments. As per reviewer suggestion, correction has been made in the revised manuscript. Please see line 118.

g. Line 180 (Page-9), R is not a language, it’s a statistical environment. Please correct.

Response- As per reviewer suggestion, correction has been made in the revised manuscript. Please see line 191.

h. Discussion section, no detail is given about the future prospects of this work.

Response- Thank you so much for your comments. As per reviewer suggestion, correction has been made. Please see lines 393-395, 410-415, 420-424 and 426-434. 

i. Language needs to be highly improved.

Response- Thank you so much for your comments and compliment. We have taken reviewer’s comment in full consideration and it will be well reflected by the revised version of manuscript. As per reviewer suggestion, correction has been made in the revised manuscript. The manuscript has now been edited by acquiring English correction services from the Mdpi Experts. Editorial Certificate is attached along with rebuttal letter. We have corrected all the grammatical mistakes in the revised manuscript. 

Reviewer #2: 

In this study, the authors predicted and comprehensively evaluated sugarcane miRNAs for the silencing of SCBV genome using in-silico algorithms. They used several tools to predict the miRNA targets and identified several potential candidate miRNAs which may be used for the silencing of SCBV. 

(Minor Concerns)

 However, the authors did not provide the expression of these miRNAs and predict their targets in the host sugarcane genome, which should be considered when selecting siRNA.

Response- Thank you so much for your comments and compliment. We have taken reviewer’s comment in full consideration and it will be well reflected by the revised version of manuscript. Yes, we have not provided the expression data in this revised manuscript. The expression of targeted miRNAs is in process.

---

## [Decision Letter · Decision Letter 1]

22 Jul 2021

PONE-D-20-35060R1

In silico genome-wide identification of Sugarcane (Saccharum officinarum L.)-encoded microRNA targets against SCBV

PLOS ONE

Dear Dr. Asharf,

Thank you for submitting your manuscript to PLOS ONE. After careful consideration, we feel that it has merit but does not fully meet PLOS ONE’s publication criteria as it currently stands. Therefore, we invite you to submit a revised version of the manuscript that addresses the points raised during the review process.

We look forward to receiving your revised manuscript.

Kind regards,

Eduardo Andrés-León

Academic Editor

PLOS ONE

Reviewers' comments:

Reviewer's Responses to Questions

**Comments to the Author**

1. If the authors have adequately addressed your comments raised in a previous round of review and you feel that this manuscript is now acceptable for publication, you may indicate that here to bypass the “Comments to the Author” section, enter your conflict of interest statement in the “Confidential to Editor” section, and submit your "Accept" recommendation.

Reviewer #2: (No Response)

2. Is the manuscript technically sound, and do the data support the conclusions?

Reviewer #2: Partly

3. Has the statistical analysis been performed appropriately and rigorously? 

Reviewer #2: Yes

4. Have the authors made all data underlying the findings in their manuscript fully available?

Reviewer #2: Yes

5. Is the manuscript presented in an intelligible fashion and written in standard English?

Reviewer #2: Yes

6. Review Comments to the Author

Reviewer #2: "author Response- Thank you so much for your comments and compliment. We have taken

reviewer’s comment in full consideration and it will be well reflected by the revised

version of manuscript. Yes, we have not provided the expression data in this revised

manuscript. The expression of targeted miRNAs is in process".

I think the authors did not address my comments.

7. PLOS authors have the option to publish the peer review history of their article (what does this mean?). If published, this will include your full peer review and any attached files.

Reviewer #2: No

---

## [Author Response · Author response to Decision Letter 1]

29 Aug 2021

Response to Reviewer’s Comments

Dear Editor,

Coauthors and I very much appreciated the encouraging, critical and constructive comments on this manuscript by the reviewer. The comments have been very thorough and useful in improving the manuscript. We strongly believe that the comments and suggestions have increased the scientific value of revised manuscript by many folds. We have taken them fully into account in revision. We are submitting the corrected manuscript with the suggestion incorporated the manuscript. The manuscript has been revised as per the comments given by the reviewers, and our responses to all the comments are as follows: 

Reviewer #2 (First Round of Revision): 

In this study, the authors predicted and comprehensively evaluated sugarcane miRNAs for the silencing of SCBV genome using in-silico algorithms. They used several tools to predict the miRNA targets and identified several potential candidate miRNAs which may be used for the silencing of SCBV. 

(Minor Concerns)

 However, the authors did not provide the expression of these miRNAs and predict their targets in the host sugarcane genome, which should be considered when selecting siRNA.

Reviewer #2 (Second Round of Revision): 

 "Author Response- Thank you so much for your comments and compliment. We have taken reviewer’s comment in full consideration and it will be well reflected by the revised

version of manuscript. Yes, we have not provided the expression data in this revised

manuscript. The expression of targeted miRNAs is in process". I think the authors did not address my comments.

Respected Sir,

We have greatly improved the existing manuscript for your kind comment after revision. It has increased the values of the current manuscript many folds. We are all very thankful to your comments.

Response- Thank you very much for your valuable suggestion. As per reviewer suggestion, correction has been made in the revised manuscript. Please see line 192-197 in the methodology section. The results section is greatly improved after revision. Please see line 349-356. The supplementary figures have been added in the manuscript (Figure S1, S2 and S3). The references section is also revised with three new references related to expression. Please see reference no. 49, 53 and 54.

---

## [Decision Letter · Decision Letter 2]

20 Sep 2021

PONE-D-20-35060R2In silico genome-wide identification of Sugarcane (Saccharum officinarum L.)-encoded microRNA targets against SCBVPLOS ONE

Dear Dr. Asharf,

Thank you for submitting your manuscript to PLOS ONE. After careful consideration, we feel that it has merit but does not fully meet PLOS ONE’s publication criteria as it currently stands. Therefore, we invite you to submit a revised version of the manuscript that addresses the points raised during the review process. Please submit your revised manuscript by Nov 04 2021 11:59PM. If you will need more time than this to complete your revisions, please reply to this message or contact the journal office at plosone@plos.org. Please include the following items when submitting your revised manuscript:A rebuttal letter that responds to each point raised by the academic editor and reviewer(s). You should upload this letter as a separate file labeled 'Response to Reviewers'.A marked-up copy of your manuscript that highlights changes made to the original version. You should upload this as a separate file labeled 'Revised Manuscript with Track Changes'.An unmarked version of your revised paper without tracked changes. You should upload this as a separate file labeled 'Manuscript'.

We look forward to receiving your revised manuscript.

Kind regards,

S.V. Ramesh, PhD

Academic Editor

PLOS ONE

Additional Editor Comments (if provided):

Although peer-reviewers see merit in your manuscript, there seems to be number of queries that remain unanswered as pointed out by one of the reviewer. Address all the comments appropriately and resubmit the manuscript for further consideration.

Reviewers' comments:

Reviewer's Responses to Questions

**Comments to the Author**

1. If the authors have adequately addressed your comments raised in a previous round of review and you feel that this manuscript is now acceptable for publication, you may indicate that here to bypass the “Comments to the Author” section, enter your conflict of interest statement in the “Confidential to Editor” section, and submit your "Accept" recommendation.

Reviewer #2: All comments have been addressed

Reviewer #3: (No Response)

2. Is the manuscript technically sound, and do the data support the conclusions?

Reviewer #2: Yes

Reviewer #3: No

3. Has the statistical analysis been performed appropriately and rigorously? 

Reviewer #2: Yes

Reviewer #3: I Don't Know

4. Have the authors made all data underlying the findings in their manuscript fully available?

Reviewer #2: Yes

Reviewer #3: Yes

5. Is the manuscript presented in an intelligible fashion and written in standard English?

Reviewer #2: Yes

Reviewer #3: No

6. Review Comments to the Author

Reviewer #2: (No Response)

Reviewer #3: The reviewers comments have been partly addressed. For example, abstract and introduction have been revised but still not up to mark. Kindly revise it. There is still no justification for the default values of parameters for prediction methods. The question "What’s the difference between a sensitive and an insensitive server?" is not answered adequately. In response to comment 8, what other confusing words have been replaced is not mentioned. Confusing/incorrect statements may be grammatically correct but technically incorrect. In the abstract the "effective badnaviral methods" is confusing, I think you mean "effective anti-badnaviral methods".

Another concern is the validation of the results, still remains.

One of the comment for "miRNAs targets in the host sugarcane genome" has not been answered too. The amiRNAs may target even the host genomes, so it is important to avoid targeting host targets which can be harmful to plants and include the ones useful to the plant in fighting the virus.

Though language corrections have been made, there is still lot of scope for further improvement.

Comments and suggestions for improvements:-

1. The revised title is grammatically incorrect. It should be “In silico identification of sugarcane (Saccharum officinarum L.) genome encoded microRNAs targeting sugarcane bacilliform virus”.

1. Why is RNAcofold used for the estimation of free energy (ΔG) associated with miRNA–mRNA interactions, while every target prediction software results give free energy value? Justify this in the manuscript.

2. In circos author used only selected sugarcane miRNAs and their SCBV targets obtained from miRanda analysis only. Please explain the criteria behind this selection of miRNAs and why only miRanda was used for this graph generation why not other tools?

3. In line number 305 it is stated that “The efficacy of the sof-miR159e target against SCBV was validated by the suppression of RNAi-mediated viral combat through the cleavage of viral mRNA or translational inhibition”. Neither any reference nor any methodology is provided regarding this validation.

4. Please justify the prediction of Consensus Secondary Structures in section 3.8 while these structures are not used anywhere to conclude anything.

5. In section 3.9, it is mentioned that table 3 comprises the free energies (ΔG) of sugarcane miRNAs and were estimated as sof-miR159 (a, b, d) (ΔG: -20.10 kcal/mol). But sof-miR159 (a, b, d) are missing from Table 3.

6. Why union plot and intersection plots are not discussed in length while main results are based on these two findings only?

7. In line 365 it is stated that “miRanda and psRNATarget are two powerful plant miRNA prediction algorithms for identifying hybridization sites in a viral genome.” The fact is that miRanda is primarily designed for the prediction of miRNA targets in animals. Please give reference in support of your assertion. Further, these algorithms are designed to predict miRNA targets rather than miRNAs.

8. In line 339 it is mentioned that the top effective candidate sof-miR159e has ΔG: -20.40 340 kcal/mol. This energy is very high for any significant hybridization of miRNA and mRNA in plants. Please justify with reference.

9. Why two highly different ΔG cut off (-15 and -20) for miRanda and RNAhybrid are used while the whole manuscript is based on the comparison.

10. How did a study of Aedes aegypti cellular miRNA and arboviruses become relevant to the present study?

11. It is mentioned that “ We have designed an amiRNA-based gene construct,”, while it is not mentioned anywhere in the method section.

12. Please include a reference for amiRNA’s mechanism of action.

13. The first-time usage of S. officinarum must be in expanded form at the first instance

14. The full-length transcript of the SCBV-BRU genome with accession no. JN377537 is used as per the manuscript, then the title of section 3.1. Genome Assembly of SCBV is misleading it should be “Genome Organization of SCBV”.

15. In line 305 it is stated that “The efficacy of the sof-miR159e target against SCBV was validated by the suppression of RNAi-mediated viral combat through the cleavage of viral mRNA or translational inhibition.” However, no reference is provided in this regard.

16. Description of tools is repetitive in the manuscript. Firstly in the method section and then in the discussion, the same lines seems redundant.

7. PLOS authors have the option to publish the peer review history of their article (what does this mean?). If published, this will include your full peer review and any attached files.

Reviewer #2: No

Reviewer #3: No

---

## [Author Response · Author response to Decision Letter 2]

19 Nov 2021

Response to Reviewer’s Comments

Dear Editor,

Coauthors and I very much appreciated the encouraging, critical and constructive comments on this manuscript by the reviewer. The comments have been very thorough and useful in improving the manuscript. We strongly believe that the comments and suggestions have increased the scientific value of revised manuscript by many folds. We have taken them fully into account in this third round of revision. We are submitting the corrected manuscript with the suggestion incorporated the manuscript. The manuscript has been revised as per the comments given by the reviewers 3 and our responses to all the comments are as follows: 

Reviewer #3 (General Comments): 

General Comment1: The reviewer’s comments have been partly addressed. For example, abstract and introduction have been revised but still not up to mark. Kindly revise it. 

Response- Thank you very much for your valuable suggestion. As per reviewer suggestion, correction has been made in the revised manuscript. Please see lines 14-23, 28-30, 40, 48-51, 62-63, 71-73, 75-77, 80, 86, 89-90 and 92-99.

General Comment2: There is still no justification for the default values of parameters for prediction methods.

Response- Thank you very much for your valuable suggestion. As per reviewer suggestion, correction has been made in the revised manuscript. Please see lines 377-381. 

General Comment3: The question "What’s the difference between a sensitive and an insensitive server?" is not answered adequately.

Response- Thank you very much for your valuable suggestion. As per reviewer suggestion, correction has been made in the revised manuscript. Please see lines 130-135. 

General Comment4: In response to comment 8, what other confusing words have been replaced is not mentioned. Confusing/incorrect statements may be grammatically correct but technically incorrect. In the abstract the "effective badnaviral methods" is confusing; I think you mean “effective ant-badnaviral methods”

Response- Thank you very much your kind comment. Correction has been made a per reviewer suggestion. Please see line 30.

General Comment5: Another concern is the validation of the results, still remains.

Response- Thank you very much for your valuable suggestion. As per reviewer suggestion, correction has been made in the revised manuscript. We agree on this concern but as this manuscript is only about screening the best possible microRNAs In silico to create virus resistance, the wet lab validation is not claimed by authors.

General Comment6: One of the comments for "miRNAs targets in the host sugarcane genome" has not been answered too. The amiRNAs may target even the host genomes, so it is important to avoid targeting host targets which can be harmful to plants and include the ones useful to the plant in fighting the virus.

Response- Thank you very much for your valuable suggestion. As per reviewer suggestion, correction has been made in the revised manuscript. The host-delivered miRNAs are retrieved from the same plant and they are already expressing inside the host. These miRNAs are expected not make much damage to the plant. We are expected that the increased in expression in the cell will stop the virus to cause extensive damage.

General Comment7: Though language corrections have been made, there is still lot of scope for further improvement:-

Response- Thank you very much for your valuable suggestion. As per reviewer suggestion, correction has been made in the revised manuscript. 

Reviewer #3 (Minor Concerns): Comments and suggestions for improvements:- 

Minor Comment: The revised title is grammatically incorrect. It should be “In silico identification of sugarcane (Saccharum officinarum L.) genome encoded microRNAs targeting sugarcane bacilliform virus”.

Response- Thank you so much for your comments and compliment. We have taken reviewer’s comment in full consideration and it will be well reflected by the revised version of manuscript. Revision has been made. Please see line: 1-2.

Comment 1: Why is RNAcofold used for the estimation of free energy (ΔG) associated with miRNA–mRNA interactions, while every target prediction software results give free energy value? Justify this in the manuscript.

Response- Thank you so much for your comments and compliment. We have taken reviewer’s comment in full consideration and it will be well reflected by the revised version of manuscript. Revision has been made. Please see lines 198-203.

Comment 2: In circos author used only selected sugarcane miRNAs and their SCBV targets obtained from miRanda analysis only. Please explain the criteria behind this selection of miRNAs and why only miRanda was used for this graph generation why not other tools?

Response- Thank you so much for your comments and compliment. We have taken reviewer’s comment in full consideration and it will be well reflected by the revised version of manuscript. Revision has been made. Please see lines 289-290 in the revised manuscript.

Comment 3: In line number 305 it is stated that “The efficacy of the sof-miR159e target against SCBV was validated by the suppression of RNAi-mediated viral combat through the cleavage of viral mRNA or translational inhibition”. Neither any reference nor any methodology is provided regarding this validation.

Response- Thank you so much for your comments and compliment. We have taken reviewer’s comment in full consideration and it will be well reflected by the revised version of manuscript. Revision has been made. Please see lines 328. Reference number 43 has been added.

Comment 4: Please justify the prediction of Consensus Secondary Structures in section 3.8 while these structures are not used anywhere to conclude anything.

Response- Thank you so much for your comments and compliment. We have taken reviewer’s comment in full consideration and it will be well reflected by the revised version of manuscript. The validation of consensual sugarcane miRNAs was confirmed by the prediction of their stable secondary structures using the RNAfold algorithm. Please see lines 335-336.

Comment 5: In section 3.9, it is mentioned that table 3 comprises the free energies (ΔG) of sugarcane miRNAs and were estimated as sof-miR159 (a, b, d) (ΔG: -20.10 kcal/mol). But sof-miR159 (a, b, d) are missing from Table 3.

Response- Thank you so much for your comments and compliment. We have taken reviewer’s comment in full consideration and it will be well reflected by the revised version of manuscript. Revision has been made. Please see table3 at line 370.

Comment 6: Why union plot and intersection plots are not discussed in length while main results are based on these two findings only?

Response- Thank you so much for your comments and compliment. We have taken reviewer’s comment in full consideration and it will be well reflected by the revised version of manuscript. Please see lines 414-421 in the discussion section.

Comment 7: In line 365 it is stated that “miRanda and psRNATarget are two powerful plant miRNA prediction algorithms for identifying hybridization sites in a viral genome.” The fact is that miRanda is primarily designed for the prediction of miRNA targets in animals. Please give reference in support of your assertion. Further, these algorithms are designed to predict miRNA targets rather than miRNAs.

Response- Thank you so much for your comments and compliment. We have taken reviewer’s comment in full consideration and it will be well reflected by the revised version of manuscript. These lines are removed. Please see lines 398-400 in the discussion section.

Comment 8: In line 339 it is mentioned that the top effective candidate sof-miR159e has ΔG: -20.40 kcal/mol. This energy is very high for any significant hybridization of miRNA and mRNA in plants. Please justify with reference.

Response- Thank you so much for your comments and compliment. We have taken reviewer’s comment in full consideration and it will be well reflected by the revised version of manuscript. The top effective candidate sof-miR159e was selected on the basis of consensual locus position predicted by at three of the algorithms used in this study.

Comment 9: Why two highly different ΔG cut off (-15 and -20) for miRanda and RNAhybrid are used while the whole manuscript is based on the comparison.

Response- Thank you so much for your comments and compliment. We have taken reviewer’s comment in full consideration and it will be well reflected by the revised version of manuscript. Please see lines 382-386.

Comment 10: How did a study of Aedes aegypti cellular miRNA and arboviruses become relevant to the present study?

Response- Thank you very much for your valuable suggestion. As per reviewer suggestion, correction has been made in the revised manuscript. This information is removed from the discussion section. Please see lines 438-440 and reference 70.

Comment 11: It is mentioned that “We have designed an amiRNA-based gene construct,” while it is not mentioned anywhere in the method section.

Response- Thank you very much for your valuable suggestion. As per reviewer suggestion, correction has been made in the revised manuscript. Please see lines 460, 462 and 463.

Comment 12: Please include a reference for amiRNA’s mechanism of action.

Response- Thank you very much for your valuable suggestion. As per reviewer suggestion, correction has been made in the revised manuscript. Please see line 75 and reference 19.

Comment 13: The first-time usage of S. officinarum must be in expanded form at the first instance.

Response- Thank you very much for your valuable suggestion. As per reviewer suggestion, correction has been made in the revised manuscript. Please see line 19.

Comment 14: The full-length transcript of the SCBV-BRU genome with accession no. JN377537 is used as per the manuscript, then the title of section 3.1. Genome Assembly of SCBV is misleading it should be “Genome Organization of SCBV”.

Response- Thank you very much for your valuable suggestion. As per reviewer suggestion, correction has been made in the revised manuscript. Please see line 214.

Comment 15: In line 305 it is stated that “The efficacy of the sof-miR159e target against SCBV was validated by the suppression of RNAi-mediated viral combat through the cleavage of viral mRNA or translational inhibition.” However, no reference is provided in this regard.

Response- Thank you very much for your comment. Correction has been made as per reviewer suggestion. Reference has been added. Please see line 328 and reference no. 43. 

Comment 16: Description of tools is repetitive in the manuscript. Firstly in the method section and then in the discussion, the same lines seems redundant.

Response- Thank you very much for your valuable suggestion. As per reviewer suggestion, correction has been made in the revised manuscript. Repetitive information and references regarding tool is removed. Please see line 379-382, 388-390 and 394-395.

---

## [Editor Report · Decision Letter 3]

23 Nov 2021

PONE-D-20-35060R3In silico identification of Sugarcane (Saccharum officinarum L.) genome encoded microRNAs targeting sugarcane bacilliform virusPLOS ONE

Dear Dr. Asharf,

Thank you for submitting your manuscript to PLOS ONE. After careful consideration, we feel that it has merit but does not fully meet PLOS ONE’s publication criteria as it currently stands. Therefore, we invite you to submit a revised version of the manuscript that addresses the points raised during the review process.

I would suggest the authors to tone down the use of amiRNA strategies etc in the MS and language needs to be rechecked once again. Please submit your revised manuscript by Jan 07 2022 11:59PM. If you will need more time than this to complete your revisions, please reply to this message or contact the journal office at plosone@plos.org. Please include the following items when submitting your revised manuscript:A rebuttal letter that responds to each point raised by the academic editor and reviewer(s). You should upload this letter as a separate file labeled 'Response to Reviewers'.A marked-up copy of your manuscript that highlights changes made to the original version. You should upload this as a separate file labeled 'Revised Manuscript with Track Changes'.An unmarked version of your revised paper without tracked changes. You should upload this as a separate file labeled 'Manuscript'.

We look forward to receiving your revised manuscript.

Kind regards,

S.V. Ramesh, PhD

Academic Editor

PLOS ONE

Additional Editor Comments (if provided):

Major issues:

This study is an in silico analysis of potential sugarcane derived miRNAs targeting Sugarcane Bacilliform Virus (SCBV) infecting the crop. The major concerns are:

a) No experimental evidence (wet lab) to prove the expression status of select or putatively antiviral miRNAs during viral infection. Had the expression data of sugarcane miRNAs was used in the analysis it would have been still acceptable. However, that was not the case.

b) Experimental evidence for the expression of host miRNAs along with RACE experiments to prove the cleavage of viral mRNA is a must to claim the antiviral role of host miRNAs. In the absence of any evidence it is a mere prediction which may or may not work.

c) When the study predicts that a select list of host miRNAs could target the viral genome sequence (or ORFs) the need for artificial miRNA construct in conferring virus resistance has not explained properly. Quite interestingly authors claim they have modified the miRNA/miRNA* sof-miRNA159. However what is the necessity for this modification or kindly explain what has been modified adequately.

d) The last sentence of the abstract states the following [The efficacies of the predicted candidate miRNAs are evaluated here to test the survival rates of the in vitro amiRNA-mediated effective anti-badnaviral methods in terms of silencing and resistance in sugarcane cultivars] however there are no experimental evidence to prove the efficacy of miRNAs in vitro or in vivo in silencing badnavirus genome.

e) A greater portion of ms is wasted in explaining the features of miRNA target prediction algorithms. Specific web links to the algorithms could suffice for the reader to understand what these softwares do.

Minor concerns:

• Still manuscript writing is not satisfactory

• There exists no connectivity whatsoever between what is presented in lines 78-82 and the lines just before (71-78).

However having said these, the title unequivocally states that the work is an in silico analysis. Hence I would suggest the authors to tone down the use of amiRNA strategies etc in the ms and language needs to be rechecked once again.
---

## [Author Response · Author response to Decision Letter 3]

23 Nov 2021

Response to Reviewer’s Comments

Dear Editor,

We are grateful to the reviewers and respected editor for their insightful comments on my papers. Coauthors and I very much appreciated the encouraging, critical and constructive comments on this manuscript by the respected editor. The comments have been very thorough and useful in improving the manuscript. We strongly believe that the comments and suggestions have increased the scientific value of revised manuscript by many folds. We have taken them fully into account in this third round of revision. We are submitting the revised manuscript with the suggestion incorporated the manuscript. We have been able to incorporate changes to reflect most of the suggestions provided by the reviewers. We have highlighted the changes within the manuscript. Here is a point-by-point response to the reviewers' comments and concerns. The manuscript has been revised as per the comments given by the respected editor and our responses to all the comments are as follows: 

Editor (Optional Comments): Major Concerns 

General Comment1: No experimental evidence (wet lab) to prove the expression status of select or putatively antiviral miRNAs during viral infection. Had the expression data of sugarcane miRNAs was used in the analysis it would have been still acceptable. However, that was not the case. 

Response- Thank you so much for your comments and compliment. We have taken reviewer’s comment in full consideration and it will be well reflected by the revised version of manuscript. Revision has been made previously and accepted by the reviewer 2. Please see lines: 193-198 and 351-358.

General Comment2: Experimental evidence for the expression of host miRNAs along with RACE experiments to prove the cleavage of viral mRNA is a must to claim the antiviral role of host miRNAs. In the absence of any evidence it is a mere prediction which may or may not work.

Response- Thank you very much for your valuable suggestion. Thank you for this suggestion. It would have been interesting to explore this aspect. However, in the case of our study, it seems slightly out of scope and is part of our research in future.

General Comment3: When the study predicts that a select list of host miRNAs could target the viral genome sequence (or ORFs) the need for artificial miRNA construct in conferring virus resistance has not explained properly. Quite interestingly authors claim they have modified the miRNA/miRNA* sof-miRNA159. However what is the necessity for this modification or kindly explain what has been modified adequately.

Response- Thank you very much for your valuable suggestion. As per reviewer suggestion, correction has been made in the revised manuscript. There is no modification. We have removed the figure 10. Please see lines 437-440 and 449-453. 

General Comment4: The last sentence of the abstract states the following [The efficacies of the predicted candidate miRNAs are evaluated here to test the survival rates of the in vitro amiRNA-mediated effective anti-badnaviral methods in terms of silencing and resistance in sugarcane cultivars] however there are no experimental evidence to prove the efficacy of miRNAs in vitro or in vivo in silencing badnavirus genome.

Response- Thank you very much for your valuable comment for the improvement of the manuscript. Correction has been made as per editor suggestion. Please see lines 28-31.

General Comment5: A greater portion of MS is wasted in explaining the features of miRNA target prediction algorithms. Specific web links to the algorithms could suffice for the reader to understand what these softwares do.

Response- Thank you very much for your valuable comment for the improvement of the manuscript. Thank you for this suggestion. It would have been interesting to explore this aspect. Appropriate changes were made and highlighted in the revised manuscript according to the suggestions of the respected editor. We agree this and have incorporated your suggestion throughout the manuscript. Please see lines 138-139, 141-143, 149-153, 157-158, 161-163,170-174, 182-183 and 185-192.

Editor Comments: Minor Concerns 

General Comment1: Still manuscript writing is not satisfactory.

Response- Thank you very much for your comments. This manuscript was edited for proper English language, grammar, punctuation, spelling, and overall style by one or more of the highly qualified native English speaking editors at MDPI Journal Experts. The editorial certificate was uploaded with the revised manuscript at first revision stage. I hope the revised manuscript will meet the requirements of academic publishing in PLoS ONE Journal and I have revised my manuscript according to the suggestions of the professional editor. Thanks again for the reminder. 

General Comment2: There exists no connectivity whatsoever between what is presented in lines 78-82 and the lines just before (71-78).

Response- The comments from the peer-reviewers are very useful for the improvement 

of our manuscript. Correction has been made in the revised manuscript. Please see lines 80-84.

General Comment3: However having said these, the title unequivocally states that the work is an in silico analysis. Hence I would suggest the authors to tone down the use of amiRNA strategies etc in the MS.

 Response- The comments from the peer-reviewers are very useful for the improvement 

of our manuscript. Correction has been made in the revised manuscript. Please see lines 94-95, 445-447 and 461.

General Comment3: Language needs to be rechecked once again.

Response- The comments from the peer-reviewers are very useful for the improvement 

of our manuscript. Thank you very much for your comments. This manuscript was edited for proper English language, grammar, punctuation, spelling, and overall style by one or more of the highly qualified native English speaking editors at MDPI Journal Experts. Thanks again for the reminder. The manuscript will be submitted again for English correction after final acceptance.

---

## [Editor Report · Decision Letter 4]

1 Dec 2021

PONE-D-20-35060R4In silico identification of Sugarcane (Saccharum officinarum L.) genome encoded microRNAs targeting sugarcane bacilliform virusPLOS ONE

Dear Dr. Asharf,

Thank you for submitting your manuscript to PLOS ONE. After careful consideration, we feel that it has merit but does not fully meet PLOS ONE’s publication criteria as it currently stands. Therefore, we invite you to submit a revised version of the manuscript that addresses the points raised during the review process. Line NO 79 Experimental validation of potential miRNA targets is highly costly and laborious.

Editor’s comments: Kindly remove this sentence.

Fig. 1 Title states “The methodology of miRNA prediction from the SCBV genome” Kindly change in to “The methodology of host or sugarcane miRNA target prediction in the SCBV genome”

Also the legend says “A flowchart designed for predicting candidate miRNAs from the SCBV genome pipeline”. In fact it is candidate miRNAs of host that could potentially target SCBV Genome. Change this too.

Kindly consult native English speaker to rectify the errors in the MS.

We look forward to receiving your revised manuscript.

Kind regards,

Shunmugiah Veluchamy Ramesh, PhD

Academic Editor

PLOS ONE

Journal Requirements:

Additional Editor Comments (if provided):

Unfortunately, I still find some errors in the manuscript. Couple of them are listed below. However, I suggest authors to thoroughly check for English usage and message they would like to convey after consulting native English speaker.

Line NO 79 Experimental validation of potential miRNA targets is highly costly and laborious.

Editor’s comments: Kindly remove this sentence.

Fig. 1 Title states “The methodology of miRNA prediction from the SCBV genome” Kindly change in to “The methodology of host or sugarcane miRNA target prediction in the SCBV genome”

Also the legend says “A flowchart designed for predicting candidate miRNAs from the SCBV genome pipeline”. In fact it is candidate miRNAs of host that could potentially target SCBV Genome. Change this too.
---

## [Author Response · Author response to Decision Letter 4]

6 Dec 2021

Response to Reviewer’s Comments

Dear Editor,

We are grateful to the reviewers and respected editor for their insightful comments on my papers. Coauthors and I very much appreciated the encouraging, critical and constructive comments on this manuscript by the respected editor. The comments have been very thorough and useful in improving the manuscript. We strongly believe that the comments and suggestions have increased the scientific value of revised manuscript by many folds. We have taken them fully into account in this fourth round of revision. We are submitting the revised manuscript with the suggestion incorporated the manuscript. We have been able to incorporate changes to reflect most of the suggestions provided by the reviewers. We have highlighted the changes within the manuscript. Here is a point-by-point response to the reviewers' comments and concerns. The manuscript has been revised as per the comments given by the respected editor and our responses to all the comments are as follows: 

Editor (Optional Comments): Minor Concerns 

General Comment1: Line NO 79 Experimental validation of potential miRNA targets is highly costly and laborious. Kindly remove it. 

Response- Thank you so much for your comments and compliment. We have taken reviewer’s comment in full consideration and it will be well reflected by the revised version of manuscript. Revision has been made. Please see line: 79.

General Comment2: Fig. 1 Title states “The methodology of miRNA prediction from the SCBV genome” Kindly change in to “The methodology of host or sugarcane miRNA target prediction in the SCBV genome”

Response- Thank you very much for your valuable suggestion. Revision has been made. Please see lines: 120-121.

General Comment3: Also the legend says “A flowchart designed for predicting candidate miRNAs from the SCBV genome pipeline”. In fact it is candidate miRNAs of host that could potentially target SCBV Genome. Change this too.

Response- Thank you very much for your valuable suggestion. As per reviewer suggestion, correction has been made in the revised manuscript. Please see lines 121-122. 

General Comment4: Kindly consult native English speaker to rectify the errors in the MS.

Response- Response- Thank you very much for your valuable comments for the improvement of our manuscript. This manuscript was edited for proper English language, grammar, punctuation, spelling, and overall style by one or more of the highly qualified native English speaking editors at MDPI Journal Experts two times: First round of revision and final round of revision. The editorial certificates were uploaded with the revised manuscript in the system. I hope the revised manuscript will meet the requirements of academic publishing in PLoS ONE Journal and I have revised my manuscript according to the suggestions of the professional editor. Appropriate changes were made and highlighted in the revised manuscript according to the suggestions of the respected English editor. We agree this and have incorporated your suggestion throughout the manuscript using service of English editor at MDPI experts. We hope the manuscript is now acceptable for publication. 

Journal Requirements: 

Response: Thank you so much for your comments and compliment. We have checked and there is no error found and all references are correct.

---

## [Editor Report · Decision Letter 5]

13 Dec 2021

In silico identification of Sugarcane (Saccharum officinarum L.) genome encoded microRNAs targeting sugarcane bacilliform virus

PONE-D-20-35060R5

Dear Dr. Asharf,

We’re pleased to inform you that your manuscript has been judged scientifically suitable for publication and will be formally accepted for publication once it meets all outstanding technical requirements.

Kind regards,

S.V. Ramesh, PhD

Academic Editor

PLOS ONE
---

## [Editor Report · Acceptance letter]

10 Jan 2022

PONE-D-20-35060R5 

*In silico* identification of Sugarcane (*Saccharum officinarum* L.) genome encoded microRNAs targeting sugarcane bacilliform virus 

Dear Dr. Asharf:

I'm pleased to inform you that your manuscript has been deemed suitable for publication in PLOS ONE. Congratulations! Your manuscript is now with our production department. 

Kind regards, 

on behalf of

Dr. Shunmugiah Veluchamy Ramesh 

Academic Editor

PLOS ONE